# Characterization of Proliferation Medium and Its Effect on Differentiation of Muscle Satellite Cells from *Larimichthys crocea* in Cultured Fish Meat Production

Shengliang Zhang [1], Hanghang Lou [1], Hongyun Lu [1], Enbo Xu [1], Donghong Liu [1,2] and Qihe Chen [1,2,*]

[1] Department of Food Science and Nutrition, Zhejiang University, Hangzhou 310058, China
[2] Innovation Center of Yangtze River Delta, Zhejiang University, Jiaxing 314100, China
* Correspondence: chenqh@zju.edu.cn

**Abstract:** To find a suitable medium for muscle satellite cells of *Larimichthys crocea*, herein, the effect of different basal media and coating materials on the proliferation of piscine satellite cells (PSCs) was explored. Firstly, two basal media, namely F10 and DMEM/F12, were selected as experimental materials, and high-sugar DMEM was the main basal culture medium used with fish muscle cells as a control. The results showed that the PSCs proliferated better in F10 than in DMEM/F12 or DMEM. Secondly, the effects of rat tail collagen, polylysine and matrix coatings, as compared with no coating, on the proliferation and later differentiation of PSCs were also investigated. Our results indicated that there was no significant difference between coating and no coating on the proliferation of PSCs in the F10-based medium. Meanwhile, it was found that the myotubes were washed out, and only those under matrix-coated conditions remained intact in the process of differentiation. The results also suggested that PSCs could still differentiate into myotubes without their stemness being affected after proliferation in the F10-based medium. Hence, this study identified an efficient proliferation medium based on F10 basal medium that could shorten the culture time and maintain the stemness of PSCs, thus providing a basis for large-scale cell expansion and cell-culture-based meat production in the future.

**Keywords:** different basal media; coating materials; proliferation; piscine satellite cells; differentiation; cell culture meat

**Key Contribution:** In this study, we obtained an efficient medium for the rapid proliferation and maintained cell stemness of muscle satellite cells from *Larimichthys crocea* for the first time; this can be used in large-scale industrial production.

## 1. Introduction

*Larimichthys crocea* (*L. crocea*), a species of the Sciaenidae, Perciformes, is one of the most commercially valuable marine fishes in China and East Asian countries. In China, the annual production of aquacultured *L. crocea* is greater than that of any other farmed marine fish species. However, frequent inbreeding, along with the rapid expansion of the *L. crocea* aquaculture industry, has resulted in the degeneration of germplasm resources and a decrease in genetic diversity, further leading to weakened disease resistance, decreased growth rates and poor meat quality in farmed *L. crocea* [1]. Cultured meat represents one of the solutions for this problem. In vitro, animal cells cultured in a medium with some growth factors can proliferate and differentiate into real meat [2].

Structurally speaking, most of the muscles of *L. crocea* belong to the white muscle of vertebrates, which is mainly composed of skeletal muscle and fat [3]. Fish skeletal muscle cell lines are often described simply by their predominate cell shape, which usually is described as either fibroblast-like, epithelial-like or spindle-like [4]. All of these cell lines can contribute to studies on in vitro meat production, and both their ante factum properties,

such as the species and anatomical site from which they arose, and post factum properties, such as growth factor requirements, are important features to consider [5]. Muscle satellite cells, which belong to the fibroblast-like cell line, have exhibited the greatest potential to differentiate into skeletal muscle cells [6]. They are generally quiescent until they are activated and become myoblasts, which undergo proliferation and myogenic differentiation to form new muscle fibers [3,7]. Cultured meat production is an efficient, safe and sustainable meat production technology [8]. In 2013, the first beef burger made from cell-cultured meat was introduced [9]. In 2019, China's first cultured pork meat was produced by Professor Zhou. Memphis meat (later known as UPSIDE Foods) made meatballs using cultured beef, chicken and duck [2,10]. Finless Foods successfully manufactured cultured meat patties using tuna stem cells. Also, with the rise and maturity of 3D technology, several researchers have successfully constructed the meat tissues of livestock such as cows [11] and pigs [12] However, few studies have used marine fish to explore in vitro myogenesis [3].

The selection of an appropriate culture medium is the key to cell culture in vitro. At present, most research focuses on the screening of cytokines or growth factors in the culture medium. However, there are few reports on the influence of different basal media on the cell culture process. In fact, basal media vary in composition and content, leading to utilization under different culture conditions. Compared with DMEM, DMEM/F12 contains more nutrients, including amino acids (L-alanine, L-aspartate and L-cysteine), vitamins (biotin and vitamin B12), inorganic salts ($CuSO_4 \cdot 5H_2O$ and $ZnSO_4 \cdot 7H_2O$), thymidine, hypoxanthine sodium and linoleic acid. Morita et al. [13] suggested that both a conditioned medium of adipose-tissue-derived stem cells (ADSCs) treated with platelet-rich plasma (PRP) and a combination of PRP with ADSC transplantation might attenuate the phosphorylation of endothelial nitric oxide synthase and angiogenesis. It was revealed by Ha et al. [14] that using asparaginase (Aspg) and glutamine synthetase (Gs) in a glutamine-free medium could enhance Chinese hamster ovary (CHO) cell productivity. Moreover, Lund et al. [15] showed that modulating the concentration of medium-chain fatty acids in culture media affected levels of the antioxidant glutathione (GSH) retained during metabolic stress in VLCAD (very long-chain acyl-CoA dehydrogenase)-deficient cell lines. Therefore, the effect of basal medium composition on cell proliferation and differentiation should be paid more attention.

Shortening culture time is very important for stem cell culture in vitro due to the fact that the differentiation potential of stem cells decreases rapidly during long-term in vitro culture, with increases in apoptosis and cell senescence [16–18]. Meanwhile, obtaining numerous stem cells is a pivotal step for cultured meat. It is essential to obtain an efficient proliferation medium to harvest more cells in less time. However, there are few reports on the optimization of proliferation media at present; most researchers have focused on suspension culture dependent on microcarriers [19,20] for large-scale expansion. Although these methods can yield enough cells, the culture cycle is still very long. In the present study, by comparing the effects of different media compositions and coating materials on cell proliferation and differentiation, the optimal culture mode for novel PSCs cells from *L. crocea* in vitro was determined. We also obtained an excellent growth medium based on F10 that could significantly shorten the culture time of PSCs and harvest more PSCs than the traditional culture medium, DMEM, while maintaining PSC stemness. Our findings could provide enough seed cells for the subsequent production of cultured meat from *L. crocea* in a short time.

## 2. Materials and Methods

### 2.1. Reagents

Dulbecco's modified eagle medium (DMEM, biosharp[®], Hefei, China), Ham's F10 (Procell[®], Wuhan, China), DMEM/F12 (Procell[®], Wuhan, China), CCK8 cell proliferation detection reagent (biosharp[®], Hefei, China), PI/RNase staining buffer (BD Pharmingen[TM], Franklin Lakes, NJ, USA), recombinant human bFGF (Beyotime[®], Shanghai, China), fetal bovine serum (FBS, WISENT[®], Nanjing, China), goat serum (Beyotime[®], Shanghai,

China), penicillin/streptomycin/amphotericin (PSA, Solarbio®, Beijing, China), ascorbic acid (aladdin®, Beijing, China), RepSox (Targetmol®, Boston, MA, USA), LY411575 (Targetmol®, Boston, MA, USA), DAPI (Solarbio®, Beijing, China), Pax7 (Bioss®, Beijing, China), MyoD1 (Abcam®, London, UK), Rb a Desmin (Bioss®, Beijing, China), Alexa Fluor 488-Labeled Goat Anti-Rabbit IgG (H+L) (Beyotime®, Shanghai, China).

### 2.2. Isolation and Culture of PSCs

All experiments relating to animals were performed according to the Chinese National Standard Laboratory Animal Guidelines for ethical review of animal welfare (GB/T 35892-2018). PSCs were isolated from the fresh muscle tissue of 6-month-old *L. crocea* seedlings purchased from the Zhoushan fish farm. Collected muscles were digested in DMEM containing 1% PSA, 0.1% collagenase IV and 4% FBS, and then in D-Hanks solution containing 0.1% trypsin. After digestion, the mixture was filtered through 70 μm and 40 μm cell filters. The isolated cells were identified with fluorescent antibodies against Pax7 (Bioss, bs-22741R) and MyoD1 (Abcam, ab209976) (Figure 1). Purified muscle satellite cells were seeded on 6-well plates and cultured in 2 mL of growth medium (GM, consisting of DMEM/F12 containing 10% FBS and 1% PSA) at 27 °C in a 5% $CO_2$ incubator. The medium was changed every two days, and when 60% confluence was reached, the passaged cells were digested using 0.25% trypsin. For long-term culture, proliferating cells were transferred to T25 and T75 flasks when the confluence of cells reached more than 80%. To induce the differentiation of PSCs, the GM was replaced with differentiation medium (DM, consisting of DMEM/F12 containing 8% FBS, 2.5 μM RepSox, 5 nM LY411575, 0.125 μM dexamethasone and 1% PSA; DM is an improvement based on the research of Xu et al. [3]) when cells reached more than 90% confluence. The differentiation process lasted for 3 to 7 days, and the medium was changed every 2 days.

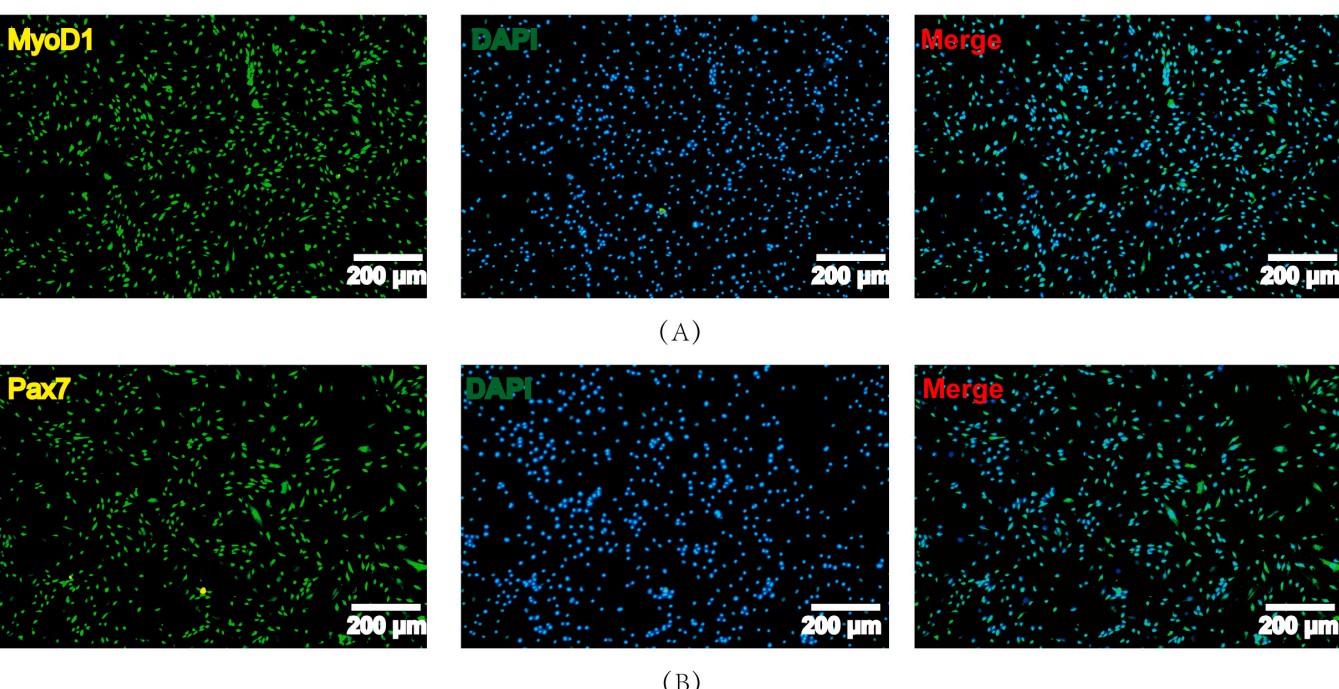

(A)

(B)

**Figure 1.** Identification of PSCs in fluorescence staining: (**A**) Green: MyoD1; Blue: DAPI; scale bar = 200 μm; (**B**) Green: Pax7; Blue: DAPI; scale bar = 200 μm.

### 2.3. CCK8 Assay

At specific time points, cell viability was determined using a CCK-8 assay. PSCs were seeded in 96-well plates ($1 \times 10^4$ cells per well) and cultured for two days. Cells were treated with CCK-8 solution for five hours, and the absorbance of each well was measured at a wavelength of 450 nm.

### 2.4. Cell Cycle Analysis

A total of $1 \times 10^6$ PSCs were seeded in a 25T culture flask and cultured for twenty-four hours. After that, cells were collected and then fixed in 70% ethanol for four hours. Then, PI/RNase staining buffer was added to samples to stain double-stranded DNA for thirty minutes. The DNA content was detected by flow cytometry and the distribution of the cell cycle was analyzed using FlowJo software (FlowJo_v10).

### 2.5. Cell Proliferation and Differentiation

At first, about $5 \times 10^4$ cells/well were prepared in cell suspension with different basal media (DMEM, F10 and DMEM/F12) and then seeded into 24-well plates coated with different materials (rat tail collagen 12 μg/mL, polylysine 0.1 mg/mL and matrix 1 mg/mL) for proliferation and culture for 3 days. After that, all the cells were replaced with a unified differentiation medium, namely DMEM + 8% FBS + 10 ng/mL bFGF + 5 μM RepSox + 10 nM LY411575 + 200 μM ascorbic acid + 1 × PSA.

### 2.6. Immunofluorescence

Cells were fixed with 4% paraformaldehyde for fifteen minutes and permeabilized with 0.3% Triton X-100 for ten minutes. Then, they were incubated with a primary antibody against desmin overnight at 4 °C. After washing with phosphate buffered saline Tween 20 (PBST), fluorescently labeled secondary antibodies were added and incubated at 37 °C for one hour, and then nuclei were stained with DAPI (4′,6-diamidino-2-phenylindole). Samples were examined using fluorescence microscopy. Nuclei counts were determined by quantifying DAPI staining using ImageJ software. The fusion index was determined by quantifying nuclei within desmin-stained myofibers as a proportion of total nuclei and multiplying by 100 [21].

### 2.7. RNA Extraction and Quantitative RT-PCR

RNA was extracted using Trizol, and then cDNA was reverse transcribed using a reverse transcription kit, after which real-time PCR was performed using SYBR Green Master Mix. All experiments were set up in triplicate, and data were analyzed using the $2^{-\Delta\Delta Ct}$ method. Primer sequences are listed in Table 1.

**Table 1.** Primer sequences for qRT-PCR.

| Gene | Primer Sequences (5′ > 3′) |
|------|------|
| *β-actin* | Forward: TGACGCGGTATAAAAGGCGA<br>Reverse: ACCAACCATCACACCCTGAT |
| *Cdk1* | Forward: GCTCTTCAGGATCTTCAGGACTCT<br>Reverse: ATTGATGACAGGTTGCCAGACTTC |
| *Cdk2* | Forward: CACGGCACCAGTATCCAGTATGA<br>Reverse: CGTCTTGAGGTCTTGGTCCACAT |

### 2.8. Statistical Analysis

Results were shown as mean ± SD. Data were analyzed using GraphPad Prism 9 software. The statistical significance of comparisons between the two groups was analyzed using multiple *t*-tests. *p* values less than 0.05 were considered statistically significant, * $p < 0.05$, ** $p < 0.01$, *** $p < 0.001$.

## 3. Results

### *3.1. Isolation and Identification of PSCs*

The isolated cells were characterized by immunofluorescence staining with specific myoblast markers such as Pax7 and MyoD1. The results demonstrated that about 99% of the cells were Pax7 positive, and 99% of the cells were MyoD1 positive (Figure 1). These data suggested that the isolated cells possessed the properties of PSCs.

### *3.2. Effect of Culture Media on PSC Proliferation*

To evaluate the effect of different basal media on the proliferation of PSCs, the cells were cultured for 18 days in uncoated 96-well plates. Meanwhile, the cell viability was determined using a CCK-8 kit at different time points. The proliferation medium consisted of different basal media (DMEM, DMEM/F12 and F10), 10% FBS, 10 ng/mL bFGF and 1% PSA. As shown in Figure 2A, in the early stage of proliferation (1–7 d), the morphology of cells cultured in DMEM/F12 and F10 basal media was thin and long, while the morphology of cells cultured in DMEM was thick and short. Then, in the late stage of proliferation (after 7 d), PSCs were all thin and long. This result suggested that different basal media could affect the development of PSCs.

With an increase in culture time, the cell density in F10 and DMEM/F12 media was significantly higher than that in DMEM (Figure 2B). As shown in Figure 2C, the total number of cells on day 18 was compared with the total number of cells on day 1 to calculate expansion fold change. The result showed that PSCs cultured with DMEM, F10 and DMEM/F12 increased in quantity by about 22-fold, 31-fold and 27-fold, respectively. F10 exhibited the best proliferation effect, followed by DMEM/F12 and DMEM. In addition, the population doubling time was calculated according to the ratio of cell viability on day 18 to that on day 1. The result showed that PSCs cultured with F10 showed the shortest population doubling time, followed by those in DMEM/F12 and DMEM (Figure 2D). Similarly, compared with that observed in the DMEM high-sugar basal medium, the expression of cell cycle regulators *Cdk1* and *Cdk2* increased about 10-fold when PSCs were cultured with F10 (Figure 2E). Also, by comparing the ratio of G0/G1 to S+G2 cells in F10 and DMEM/F12 media, the value seen in F10 media was less than that in DMEM/F12. This result demonstrated that F10 was more effective than DMEM/F12 for promoting cell proliferation (Figure 2F).

In conclusion, F10 or DMEM/F12 could significantly better promote the long-term proliferation of PSCs compared to DMEM. Similarly, F10 had a more potent effect than DMEM/F12.

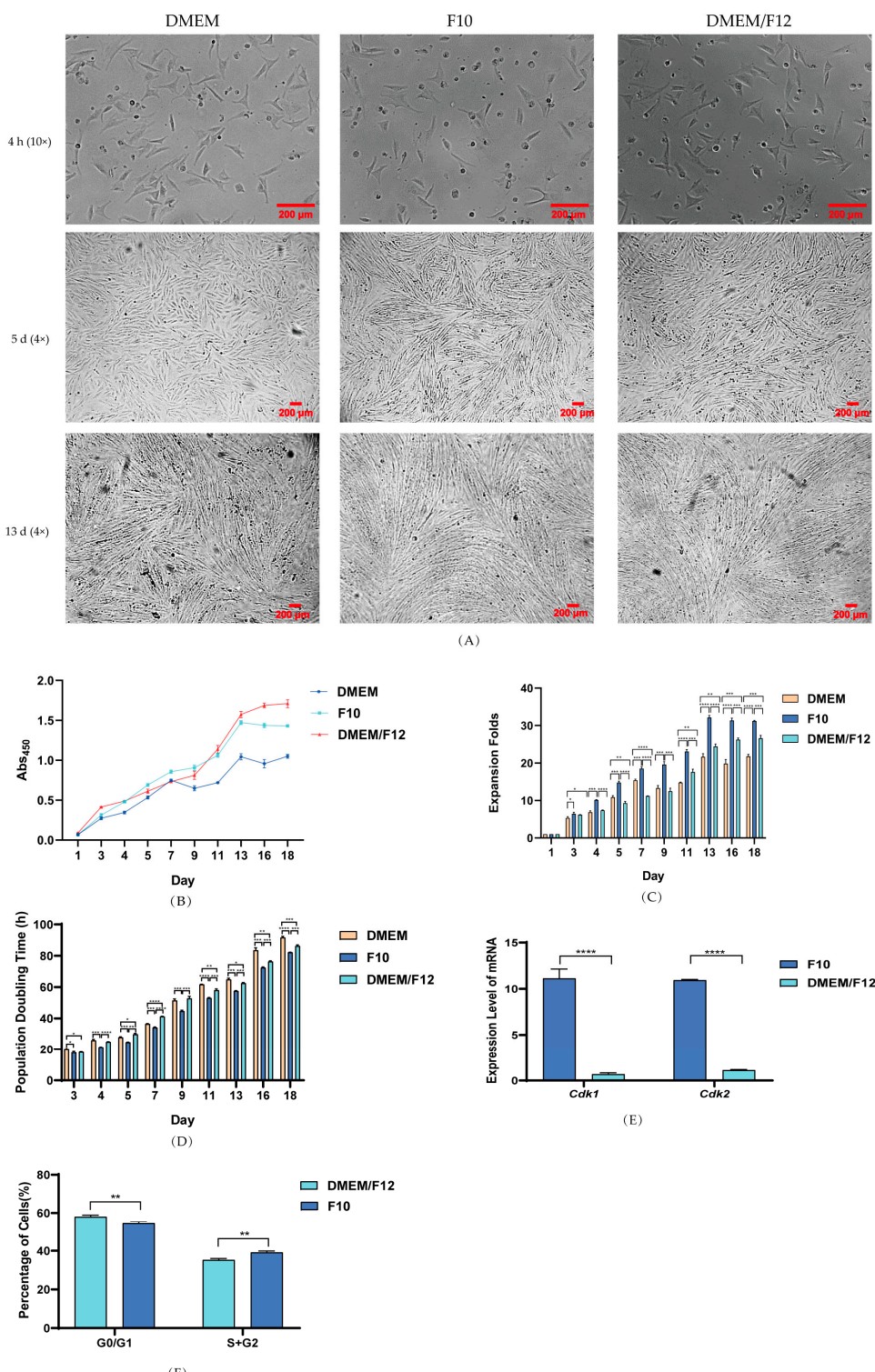

**Figure 2.** F10 and DMEM/F12 basal media significantly promoted PSC proliferation: (**A**) Typical electron microscopic images of PSCs cultured in different basal media, scale bar = 200 μm; (**B**) plot of PSC growth under different basal media; (**C**) proliferation fold changes for PSCs treated with different basal media at different time points; (**D**) population doubling time of PSCs in different basal media at different time points; (**E**) quantification of *Cdk1* and *Cdk2* mRNA expression (DMEM high-sugar basal medium as a control); (**F**) cell cycle analysis of PSCs at day 1 by flow cytometry. The results are presented as mean ± SD. All the experiments were repeated at least three times. * $p < 0.05$, ** $p < 0.01$, *** $p < 0.001$, **** $p < 0.0001$.

### 3.3. Effect of Coated and Uncoated Materials on PSC Proliferation

According to the above experimental results, both F10 and DMEM/F12 could significantly promote PSC proliferation under uncoated conditions. At present, most researchers have coated the in vitro culture vessels. Herein, three commonly used coating materials (matrix, rat tail collagen and polylysine) were used to investigate the effects of different basal media on PSC proliferation for 5 days. On the fifth day of culture, the morphology of PSCs cultured in F10 and DMEM/F12 basal media under both coated and uncoated conditions showed a slender, normal morphology (Figure 3A). With an increase in culture time, the cell viability of PSCs in DMEM/F12 and F10 media increased significantly under both coated and uncoated conditions (Figure 3B). As shown in Figure 3C, the total number of cells on day 5 was compared with the total number of cells on day 1 to calculate the expansion fold change. In the F10-based medium, the quantity of PSCs cultured in uncoated, rat-tail-collagen-coated, polylysine-coated and matrix-coated vessels increased by about 42-, 19-, 26- and 41-fold, respectively. The result showed that there was no significant difference in PSC expansion between the uncoated group and the matrix-coated group ($p > 0.05$). In the DMEM/F12-based medium, the PSC expansion fold change in uncoated, rat-tail-collagen-coated, polylysine-coated and matrix-coated vessels increased by about 21-, 17-, 19- and 24-fold, respectively. These results revealed that the PSCs in the matrix-coated group exhibited the best expansion, followed by those in the uncoated group, with a significant difference between the two ($p < 0.001$). To sum up, the effect of F10 was better than that of DMEM/F12 with or without coating, which was possibly related to the different composition.

In addition, as shown in Figure 3D, the population doubling time was calculated according to the ratio of cell viability on day 5 to that on day 1. In the F10-based medium, PSCs cultured under uncoated, rat-tail-collagen-coated, polylysine-coated and matrix-coated conditions required about 18, 23, 20 and 18 h, respectively, for doubling, suggesting that both uncoated and matrix-coated conditions contributed to the shortest population doubling times ($p > 0.05$). In the DMEM/F12-based medium, PSCs cultured under uncoated, rat-tail-collagen-coated, polylysine-coated and matrix-coated conditions required about 22, 24, 23 and 21 h, respectively, for doubling. This result showed that uncoated conditions were the best, followed by matrix-coated conditions, with no significant difference between the two ($p > 0.05$). Generally speaking, the population doubling time in F10 medium was shorter than that in DMEM/F12 under both coated and uncoated conditions during the same culture period, indicating that cells grew faster in the F10-based medium.

In conclusion, the F10-based medium was more conducive to cell proliferation. Additionally, whether PSCs were cultured in F10 or DMEM/F12, the matrix coating yielded the best proliferation effect among the three coating materials. Meanwhile, there was no significant difference in proliferation between matrix-coated and uncoated conditions in F10-based medium, but in the DMEM/F12-based medium, there was a slight difference. These findings suggested that differences in medium composition could give rise to differences in cell proliferation on coated materials.

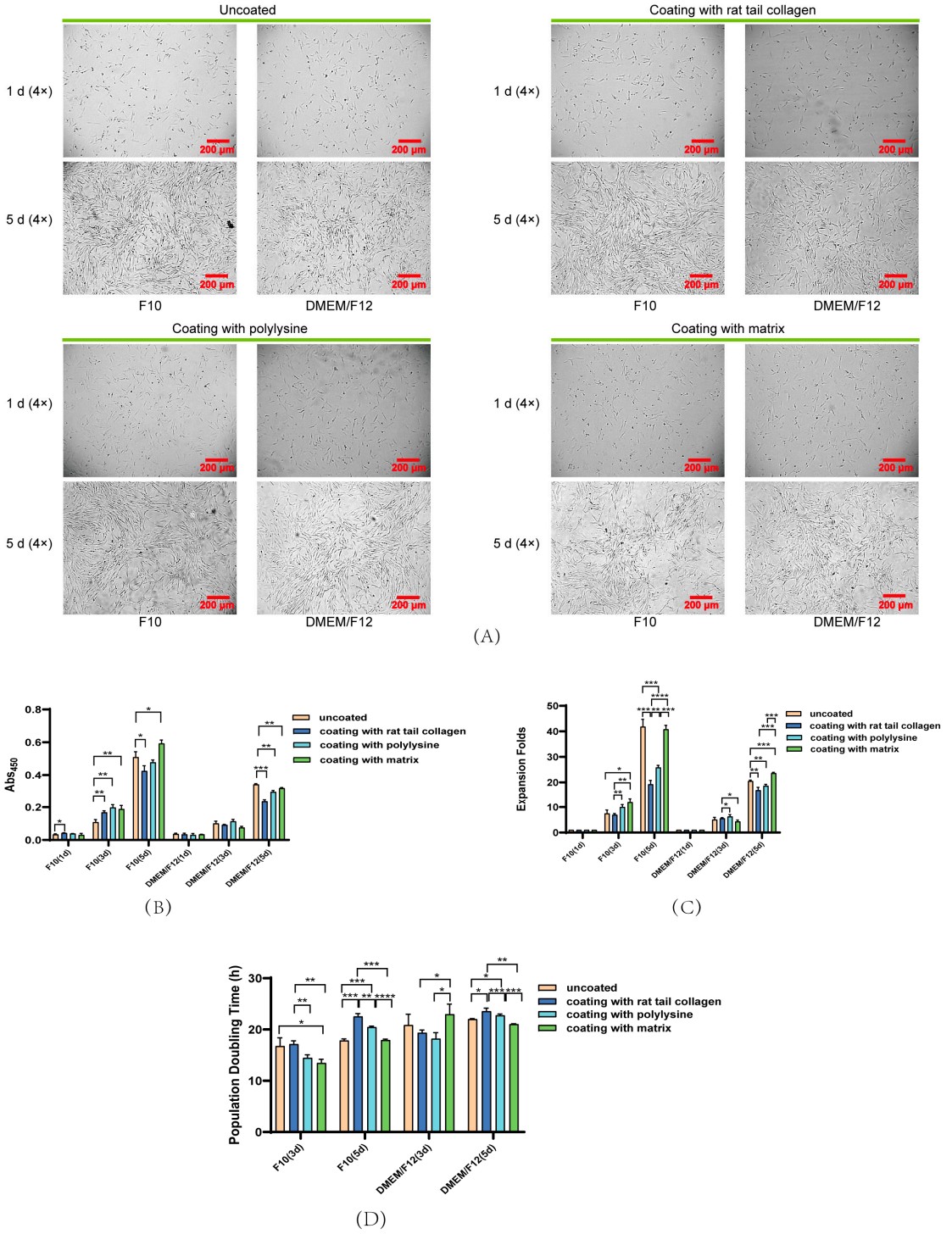

**Figure 3.** Matrix coating promoted PSC proliferation: (**A**) Typical electron microscopic images of PSCs cultured in different basal media and different coating materials, scale bar = 200 μm; (**B**) relative cell viability, determined by CCK-8 assay, for PSCs after incubation with different basal media and different coating materials for 5 d; (**C**) proliferation fold change in PSCs treated with different basal media and different coating materials at different time points; (**D**) population doubling time of PSCs in different basal media with different coating materials at different time points. The results are presented as mean ± SD. All the experiments were repeated at least three times. * $p < 0.05$, ** $p < 0.01$, *** $p < 0.001$, **** $p < 0.0001$.

### 3.4. Effects of Basal Media and Coated Materials on PSC Proliferation

After investigating the effects of different basal media and coating materials on the proliferation of muscle satellite cells, the influence of these materials on the differentiation of muscle satellite cells was also explored. During the proliferation process, the cells cultured in DMEM basal medium did not grow. The formation of myotubes under different conditions was observed by microscopy and electron microscopy. No matter whether the basal growth medium was DMEM/F12 or F10, more myotubes formed and the differentiation effect of cells was better under matrix-coated conditions (Figure 4). In the process of fluorescence staining, it was found that the rest of the myotubes were washed out, and only the ones coated with matrix glue were intact, which might have been caused by the strong adhesion properties of the matrix. Under the matrix-coated conditions, it can be seen from Figure 5A that the differentiation effect of F10 was better in immunofluorescence staining, and the fusion index reached about 20%, while that in DMEM/F12 was only about 10%. To sum up, the proliferation of PSCs in the growth medium based on DMEM/F12 or F10 did not affect their later differentiation, and PSCs cultured in F10 media formed more myotubes.

**Figure 4.** *Cont.*

GM:F10

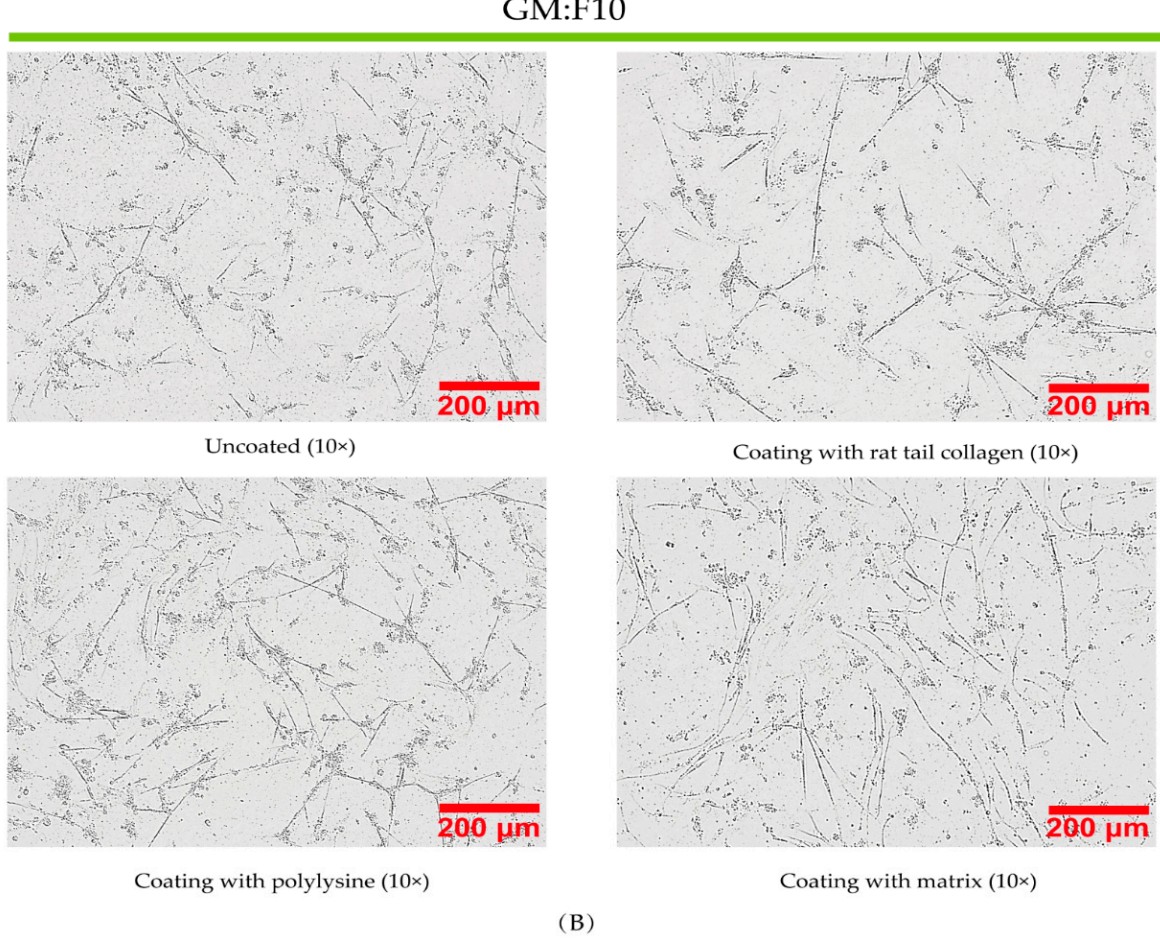

(B)

**Figure 4.** Effect of different coating materials on myoblast differentiation, scale bar = 200 μm: (**A**) The basal medium for proliferation was DMEM/F12; (**B**) the basal medium for proliferation was F10.

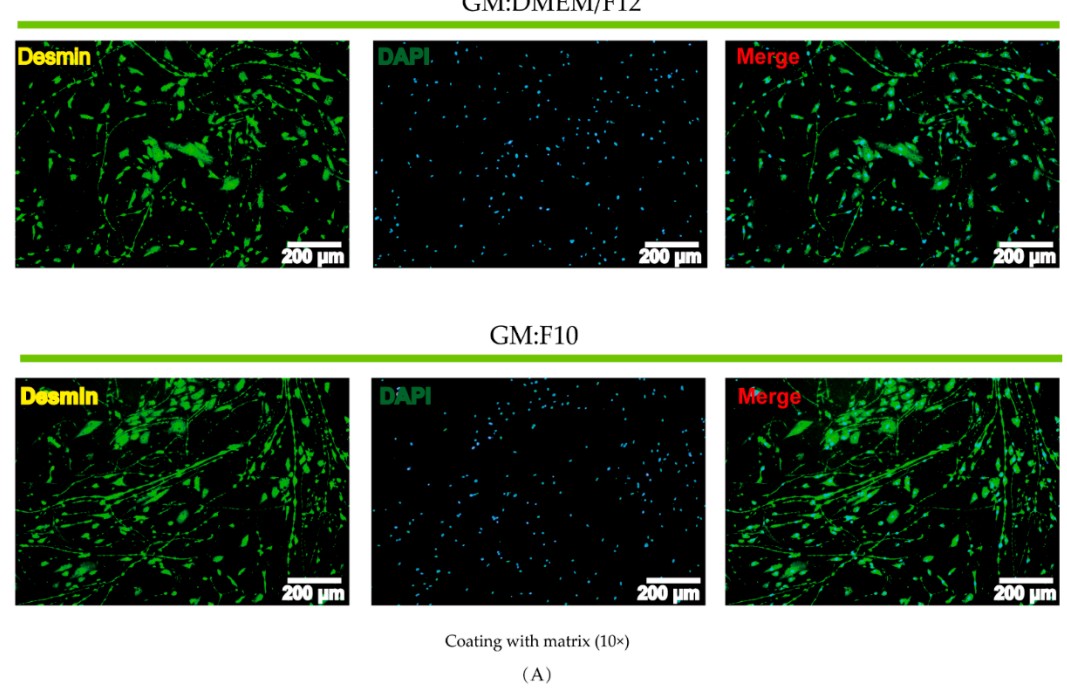

Coating with matrix (10×)

(A)

**Figure 5.** *Cont.*

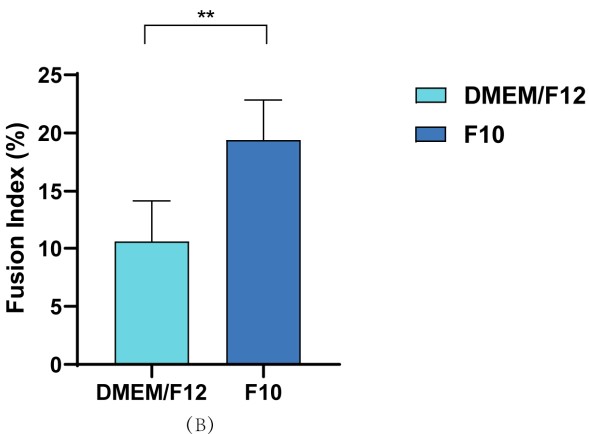

(B)

**Figure 5.** Matrix coating promoted PSC proliferation: (**A**) Representative images of fluorescence staining of PSCs during myogenic differentiation at day 4. Green: desmin; Blue: DAPI; scale bar = 200 μm; (**B**) fusion index of PSC growth with matrix coating and different basal media. The results are presented as mean ± SD. All the experiments were repeated at least three times. ** $p < 0.01$.

## 4. Discussion

It is essential to obtain an efficient proliferation medium to shorten cell culture time and produce as many cells as possible within a limited time. Such an approach would be very beneficial for the industrialized production of cultured meat. Some researchers have reported that differentiation potential decreases rapidly while apoptosis and cell senescence increase during long-term in vitro culture. For instance, it was reported by Wagner et al. that mesenchymal stem cell differentiation potential declined and senescence-associated gene expression increased over time [22]. Qingzi lei et al. also found that the decline in differentiation potential of porcine muscle stem cells was related to culture time rather than cell division number before differentiation [23]. In the previous exploration of proliferation media for PSCs, it was found that cell viability and growth rates decreased when PSCs grew to a certain number of generations in a DMEM-based medium. Interestingly, the cell viability and growth rate increased when the basal medium was replaced with F10 instead of DMEM. This result demonstrated that different basal media have very different effects on the cells, so two basal media, namely F10 and DMEM/F12, were chosen to explore the effects of media on the growth of muscle satellite cells. Herein, we obtained an efficient proliferation medium based on F10 basal medium that effectively promoted the proliferation of muscle satellite cells from *L. crocea* in vitro for the first time. By analyzing the expansion fold change, population doubling time, expression of cell proliferation marker genes *Cdk1* and *Cdk2* and myogenic differentiation potential, we demonstrated that the effect of F10 was the best. The total cell number of PSCs cultured with DMEM, F10 and DMEM/F12 increased by about 22-fold, 31-fold and 27-fold, respectively, and the proliferation of PSCs in the F10-based medium did not affect later differentiation. There were more myotubes that formed in the F10-based medium, and the fusion index reached about 20%.

There is no denying that different basal media have diverse functional effects. Related studies have shown that DMEM was used for cell proliferation, whereas DMEM/F12 was used for cell differentiation and Ham's F10, which contained nutrients such as vitamins, amino acids and metabolites, was suitable for long-term, serum-free cultures [24]. Basal media vary in their compositions and contents, leading to their utilization under different culture conditions. Overall, the concentrations of amino acids and vitamins in DMEM were much greater than those in DMEM/F12 and F10. Fayaz and Honaramooz [25] showed that DMEM sufficiently supported the in vitro propagation of gonocytes and somatic cells. Pahlavanneshan et al. [26] demonstrated that Ham's F10 medium provided better conditions for the 2D culture of primary islet cells. Hua et al. [27] showed that

DMEM/F12 medium was the most suitable medium for lower-serum adherent culture among four different media (F12, DMEM/F12, 1640 and DMEM). Wei et al. [28] indicated that glutamine was the key molecule for maintaining cell growth and survival in culture. In this study, DMEM and DMEM/F12 contained 4 and 2.5 mM glutamine, respectively, and F10 contained 1.5 mM alanyl-glutamine. This composition might be a key factor in promoting PSC proliferation. Singh et al. [19] investigated the effects on cell growth by comparing different components of DMEM/F12 medium with DMEM, which supported our findings in PSCs.

Cell adhesion and migration are essential for cell proliferation, and some researchers have changed the structure of cell contact media to promote cell adhesion. Du et al. [29] grafted silk sericin onto the surface of a thermoplastic polyurethane (TPU) membrane by utilizing a $NH_2$ bridge to build a high-efficiency cell culture membrane that could promote cell adhesion. Dhania et al. [30] showed that different cells exhibited a significant increase in cell viability and attachment when cultured on the porous mats of fabricated polyhydroxyalkanoate-blend scaffolds. In this study, we adopted the latter approach by applying a coating material to the cell contact medium to promote cell attachment. Three common coating materials (rat tail collagen, matrix and polylysine) were selected. Among them, matrix is widely used to mimic the environment of the extracellular matrix (ECM), which assists in maintaining the optimal balance for a series of cell adjustment behaviors, including cell proliferation, migration and differentiation [31]. The ECM plays a major role in cell–cell and cell–matrix signaling during normal physiology and disease [32]. The results showed that there was no significant difference in the proliferation of PSCs between coated and uncoated vessels, and the effect of matrix was the best among the coating materials. The reason for this phenomenon might be related to species or cell types, and it was indicated that the nutritional components of these three basic media could meet the growth of PSCs. There is no need to provide the culture with additional external adhesion or migration forces.

## 5. Conclusions

As we all know, obtaining enough stem cells in vitro is a pivotal step for producing cultured meat. For this reason, this study compared the effects of different basal media on the proliferation and later differentiation of PSCs. Finally, an efficient proliferation medium based on F10 basal medium was obtained. The total quantity of PSCs cultured with F10 medium increased about 31-fold, a significantly higher increase than that seen for the traditional culture medium DMEM (22-fold). As the same time, in order to promote cell adhesion and migration, three common coating materials (rat tail collagen, matrix and polylysine) were selected. Our results suggested that there was no significant difference between coated and uncoated substrates regarding the proliferation of PSCs cultured in F10 medium. It was also found that myotubes were washed out and only the ones coated with matrix were intact in the process of differentiation, which might be caused by the strong adhesion ability of matrix. Under the matrix-coated conditions, the fusion index of F10 reached about 20%. It was indicated that the proliferation of PSCs in the F10-based medium did not affect later differentiation. Overall, we obtained an excellent proliferation medium that allowed for rapid proliferation and maintained stemness of PSCs. Thus, our findings will provide a basis for large-scale cell expansion and cell culture meat production in the future.

**Author Contributions:** Conceptualization, Q.C.; methodology, S.Z.; software, S.Z.; validation, H.L. (Hanghang Lou), H.L. (Hongyun Lu) and E.X.; formal analysis, S.Z. and H.L. (Hanghang Lou); investigation, S.Z.; resources, Q.C. and D.L.; data curation, H.L. (Hanghang Lou); writing—original draft preparation, S.Z.; writing—review and editing, H.L. (Hongyun Lu); visualization, S.Z.; supervision, Q.C.; project administration, Q.C. and D.L.; funding acquisition, Q.C. and D.L. All authors have read and agreed to the published version of the manuscript.

**Funding:** This research was funded by the Starry Night Science Fund of Zhejiang University Shanghai Institute for Advanced Study (Grant No. SN-ZJU-SIAS-004).

**Institutional Review Board Statement:** Not applicable.

**Informed Consent Statement:** Not applicable.

**Data Availability Statement:** The data that support the findings of this study are available upon request from the corresponding author.

**Conflicts of Interest:** The authors declare no conflict of interest.

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
