# Peer review of "Characterization of Proliferation Medium and Its Effect on Differentiation of Muscle Satellite Cells from Larimichthys crocea in Cultured Fish Meat Production"

_fishes, doi:10.3390/fishes8090429_

Round 1

Reviewer 1 Report

The manuscript sounds relevant and brings interesting analysis of cell culture media to improve establishment and production of in vitro muscle cells from Larimichthys crocea, with interesting findings on cellular viability, proliferation and differentiation. However, it has some major issues, which need revision and/or correction before publication.

MAJOR:

Although the work shows relevance, good experimental design and adequate techniques, it seems that the manuscript was written without much revision. Poor introduction, references out of context, sentences bad written, described methods that were not used in the study, text errors...

Results: In section 3.3, the authors wrote “No matter whether the basic growth medium was DMEM/F12 or F10, more myotubes could be formed, (…)”.

To state that "more myotubes were formed", it would be important to show the differentiation results of DMEM basal medium. Since the study focus on the improved effect of F10 and DMEM/F12 growth media on cell proliferation and differentiation, we have to see that the DMEM had lower effects.

Discussion focused only at the selected culture media and coating materials. There is almost none discussion about the results from cell viability, proliferation and differentiation. The authors should provide other studies that evaluated fish muscle cells in vitro, discussing the used medium and the outcomes.

MINOR:

Abstract: “(…) as experimental materials, namely DMEM high sugar, F10 and DMEM/F12 (…)”.

To give more strength and relevance to the results, I suggest indicate that the "DMEM with high sugar" is the main culture medium used for fish muscle cell cultures.

Abstract: “This study obtained a proliferation medium based on DMEM/F12 or F10 can make the rapid proliferation of PSCs in a short time, (…)”.

Review this sentence.

Key contribution: “(…) and maintained cell dryness for the first time(…)”.

Considering the claim that cell dryness was observed for the first time, it is important to include and describe this result throughout the manuscript.

Introduction: “How to solve this problem, cultured meat is one of the solutions. In vitro, animal cells cultured with medium and some growth factors to proliferate and differentiate into real meat. Compared witwh the potential of several cell types for the production of cultured meat and find out that muscle satellite cells have exhibited the greatest potential which can differentiate into skeletal muscle cells.”

Review this paragraph.

Introduction: The authors focused only on the importance of in vitro meat production. It is important to highlight also that improved proliferation medium could contribute to studies and research aiming to understand the fish muscle biology and growth.

Introduction: “So far, there is no relevant report on the study of different basal media on the proliferation of the same kind of cells. In the case of the same other components of the medium, choosing different basic media, the results of the effects on proliferation rate, cell morphology and stemness of PSCs were quite different”.

Very confusing. Review this paragraph.

Introduction: I did not understand the references cited here. None of them used fish as the experimental models or even are related to muscle. I suggest to check for other references, more linked to the present study. The same for the Discussion section.

Introduction: Specify the meaning of abbreviations (CHO, GSH, VLCAD).

Materials and Methods: “The isolated cells were identified with fluorescent antibodies against Pax7 (Bioss, bs-22741R) and MyoD1 (Abcam, ab209976).”

Where are the results of Pax7 and Myod1 immunofluorescence? They are necessary.

Materials and Methods: “To induce differentiation of PSCs, GM was replaced with differentiation medium (DM, consisting of DMEM/F12 containing 4%FBS, 2.5uM repsox, 5nM LY411575, 0.125 uM dexamethasone, and 1%PSA) when cells reached more than 90% confluence”.

It is not clear why this differentiation media (DMEM/F12, 4% FBS, repsox, LY411575, dexamethasone, 1% PSA) was used. What is the difference between this differentiation induction from the one explained in 2.5 section ("Cell differentiation")?

Materials and Methods: “(…) 24-well plates coated with different materials (rat tail collagen, polylysine and Matrigel) (…)”.

How this coating materials were selected? With what bases? One of the most used coating for fish muscle cell cultures is the combination of poly-lysine and laminin, which show high affinity for myoblast (and possibly could prevent the wash out seen at the immunofluorescence). I wonder why does the laminin was not selected in the study?

Materials and Methods: “Statistical significance of comparisons between the two groups was analyzed using multiple t-tests”.

Which two groups? There are 3 different media (DMEM, F10 and DMEM/F12) and 3 different coating materials (collagen, polylysine and matrigel). Why did not used ANOVA (parametric) or Kruskall-Wallis (non-parametric)?

Results: The numbers of the figures indicated in the text do not correspond to the images in the manuscript. Review all of them.

Results: “In conclusion, F10 and DMEM/F12 could significantly promote the long-term proliferation of PSCs by regulating cell cycle and enhancing cell proliferative activity”.

I also suggest indicate that F10 has a more potent effect, compared to DMEM/F12.

Results: In the Figure 1 (wrongly indicated as figure 2), the quality of the images could be better.

Results: In section 3.2, the end of the first paragraph and the beginning of the second paragraph are very repetitive. It is necessary to synthesize the information.

Results: “(…) and the fusion index reached about 20%, while that of DMEM/F12 was only about 10% (Fig. 4A and 4B)”.

Explain in the "Methods" section how the fusion index was analyzed and obtained.

Discussion: “Alanyl-glutamine is very stable in aqueous solution and does not degrade spontaneously. This composition might be an underlying mechanism for its great support for cell propagation and colony-formation in the present study”.

In addition, F10 and DMEM/F12 media could not show better results due to their lower concentration of glutamine compared to DMEM?

Discussion: The last paragraphs on page 10 are poorly written.

Discussion: “In this study, the concentrations of amino acids and vitamins in DMEM were much more abundant than in DMEM/F12 and F10”.

Considering the higher amount of amino acid and vitamins in DMEM, the authors should discuss better how this was the medium with the less promising results.

Discussion: This section needs to be further explored. How the achievements of the work will be important to meat production, largely cited in the introduction?

The quality of English is moderate. Authors need to proofread the text, especially the concordance and the link between sentences/paragraphs. Many sentences are confusing and/or out of context.

Author Response

Responses to comments of Editor

Thank you for your serious and constructive comments on our manuscript. According to your suggestion, the manuscript has been revised as a letter to editor. The revisions we have made are as follows:

Ø 1.1 The first reviewer's comments:Results: In section 3.3, the authors wrote “No matter whether the basic growth medium was DMEM/F12 or F10, more myotubes could be formed, (…)”. To state that "more myotubes were formed", it would be important to show the differentiation results of DMEM basal medium. Since the study focus on the improved effect of F10 and DMEM/F12 growth media on cell proliferation and differentiation, we have to see that the DMEM had lower effects.

Response: Thank you for your constructive and helpful suggestion. The reason why there was no data on cell differentiation under the effect of DMEM basal medium was that DMEM/F12 basal medium or F10 basal medium had a better effect on cell proliferation when we discussed the influence of different basal medium on cell proliferation in the early stage, so we always used DMEM/F12 basal medium for cell culture. It may be due to the domestication of cells, the cells did not grow after being changed to DMEM medium, so it is impossible to study the effect of DMEM on cell differentiation. the cell growth in DMEM basal medium Image is shown as below:              The reason why we say that DMEM/F12 or F10 will form more myotubes is because the cells had previously been cultured with DMEM, during this period, cells can grow and differentiate normally, and we can get the data of differentiation. We compared the DMEM data at this time with the DMEM/F12 and F10 data later, and we found that DMEM/F12 or F10 could form more myotubes.I hope my answer has solved your doubts and made you satisfied. Thank you.

Ø 1.2 The first reviewer's comments:Discussion focused only at the selected culture media and coating materials. There is almost none discussion about the results from cell viability, proliferation and differentiation. The authors should provide other studies that evaluated fish muscle cells in vitro, discussing the used medium and the outcomes.

Response: Thank you for your constructive and helpful suggestion. The “Discussion” section has been rewritten as you suggested. This section is briefly described as below: The first paragraph discussed the results from cell viability, proliferation and differentiation; The second paragraph mainly explained that the longer the time of cell culture in vitro, the cell differentiation ability and vitality will decline, so it is necessary to obtain a high-efficiency proliferation medium; Which leads to the third paragraph, the third paragraph is about exploring an efficient proliferation medium from different basal medium; In order to further improve the cell proliferation efficiency, consider the use of coated materials, the fourth paragraph describes the coating material. Please see the section 4 “Discussion” of the manuscript. I hope my modification has solved your doubts and made you satisfied. Thank you.

Ø 1.3 The first reviewer's comments:Abstract: “(…) as experimental materials, namely DMEM high sugar, F10 and DMEM/F12 (…)”. To give more strength and relevance to the results, I suggest indicate that the "DMEM with high sugar" is the main culture medium used for fish muscle cell cultures.

Response: Thank you for your constructive and helpful suggestion The sentence has been changed to “Firstly, two basal media namely F10 and DMEM/F12 were selected as experimental materials and DMEM high sugar was the main culture medium used for fish muscle cell cultures as a control”. Please see highlight in the “Abstract” section. I hope my modification has solved your doubts and made you satisfied. Thank you.

Ø 1.4 The first reviewer's comments:Abstract: “This study obtained a proliferation medium based on DMEM/F12 or F10 can make the rapid proliferation of PSCs in a short time, (…)”. Review this sentence.

Response: Thank you for your constructive and helpful suggestion. The sentence has been changed to “This study obtained an efficient proliferation medium based on DMEM/F12 or F10. It could shorten the culture time and maintain the stemness of PSCs, providing a basis for large-scale cell expansion and cell culture meat production in the future”. Please see highlights in the “Abstract” section.

Ø 1.5 The first reviewer's comments:Key contribution: “(…) and maintained cell dryness for the first time(…)”. Considering the claim that cell dryness was observed for the first time, it is important to include and describe this result throughout the manuscript.

Response: Thank you for your constructive and helpful suggestion. Firstly, I'm sorry about the spelling mistakes, “dryness” should be changed to “stemness”. Secondly, “the first time” refer to this is the first time to study the culture of muscle satellite cells of Larimichthys crocea in vitro at home and abroad. The sentence has been changed to “In this study, we obtained an efficient medium for the rapid proliferation and maintained cell stemness of muscle satellite cells from Larimichthys crocea for the first time, which can be used in large-scale industrial production”. Please see highlights in the “Key Contribution” section.

Ø 1.6 The first reviewer's comments:Introduction: “How to solve this problem, cultured meat is one of the solutions. In vitro, animal cells cultured with medium and some growth factors to proliferate and differentiate into real meat. Compared with the potential of several cell types for the production of cultured meat and find out that muscle satellite cells have exhibited the greatest potential which can differentiate into skeletal muscle cells.” Review this paragraph.

Response: Thank you for your constructive and helpful suggestion. The paragraph has been changed to “How to solve this problem, cultured meat is one of the solutions. In vitro, animal cells cultured with medium and some growth factors to proliferate and differentiate into real meat. Real meat is mostly made up of muscle cells and fat cells. Muscle satellite cells have exhibited the greatest potential which can differentiate into skeletal muscle cells.” Please see highlights in the “Introduction” section.

Ø 1.7 The first reviewer's comments:Introduction: The authors focused only on the importance of in vitro meat production. It is important to highlight also that improved proliferation medium could contribute to studies and research aiming to understand the fish muscle biology and growth.

Response: Thank you for your constructive and helpful suggestion. We searched the keywords "basic medium" and "cell proliferation" on the Web of Science and found that there were very few related literature reports. I think your suggestions are very important and we will explore the effect of the improved proliferation medium on the fish muscle biology and growth from the mechanism. At present, the purpose of this paper is to provide a large number of seed cells for our team's later cultured fish meat, which requires the acquisition of an efficient proliferation medium, so we mainly focused on the in vitro meat production.

Ø 1.8 The first reviewer's comments:Introduction: “So far, there is no relevant report on the study of different basal media on the proliferation of the same kind of cells. In the case of the same other components of the medium, choosing different basic media, the results of the effects on proliferation rate, cell morphology and stemness of PSCs were quite different”. Very confusing. Review this paragraph.

Response: Thank you for your constructive and helpful suggestion. The paragraph has been changed to “In this study, we found that the selection of different basal media had a great difference in proliferation rate, cell morphology and stemness of PSCs.” Please see highlights in the “Introduction” section.

Ø 1.9 The first reviewer's comments: Introduction: I did not understand the references cited here. None of them used fish as the experimental models or even are related to muscle. I suggest to check for other references, more linked to the present study. The same for the Discussion section.

Response: Thank you for your constructive and helpful suggestion. We searched the keywords "basic medium" and "cell proliferation" on the Web of Science and found that there were very few related literature reports, so we selected several relevant literatures that could explain some problems that talking about different basal media. The “Discussion” section has been rewritten as you suggested. Please see the section 4 “Discussion” of the manuscript. The penultimate paragraph of the part of introduction has been changed to “Selection of appropriate culture medium is crucial for cell culture in vitro. Different basal media are used in different culture conditions because of their different components and content. In this study, we found that the selection of different basal medium had a great difference in proliferation rate, cell morphology and stemness of PSCs. There is no relevant report on the study of different basal media on the cell proliferation. At present, most research focuses on the screening of cytokines or growth factors in culture medium. Ha, et al. [8] have revealed that using asparaginase (Aspg) and glutamine synthetase (Gs) in glutamine free medium could enhance CHO cell productivity. Lund, et al. [9] have showed that modulation of the concentration of medium-chain fatty acids in culture media affects levels of GSH retained during metabolic stress in VLCAD-deficient cell lines.”

Ø 1.10 The first reviewer's comments:Introduction: Specify the meaning of abbreviations (CHO, GSH, VLCAD).

Response: Thank you for your constructive and helpful suggestion. The full name of abbreviations as below and has been supplement with highlight in the article. CHO: Chinese hamster ovary; GSH: the antioxidant glutathione; VLCAD: very long-chain acyl-CoA dehydrogenase

Ø 1.11 The first reviewer's comments:Materials and Methods: “The isolated cells were identified with fluorescent antibodies against Pax7 (Bioss, bs-22741R) and MyoD1 (Abcam, ab209976).” Where are the results of Pax7 and Myod1 immunofluorescence? They are necessary.

Response: Thank you for your constructive and helpful suggestion. We have already added in the text. Please see Fig.1 in the manuscript.

Ø 1.12 The first reviewer's comments:Materials and Methods: “To induce differentiation of PSCs, GM was replaced with differentiation medium (DM, consisting of DMEM/F12 containing 4%FBS, 2.5uM repsox, 5nM LY411575, 0.125 uM dexamethasone, and 1%PSA) when cells reached more than 90% confluence”. It is not clear why this differentiation media (DMEM/F12, 4% FBS, repsox, LY411575, dexamethasone, 1% PSA) was used. What is the difference between this differentiation induction from the one explained in 2.5 section ("Cell differentiation")?

Response: Thank you for your constructive and helpful suggestion. The differentiation medium mentioned in this study is an improvement based on the research of our team through transcriptomic exploration in the previous stage and all the differentiation medium formulations mentioned in the article are the same. The article has been published (Tissue-like cultured fish fillets through a synthetic food pipeline. NPJ Sci Food 2023, 7, 17, doi:10.1038/s41538-023-00194-2.). Please see the highlight in section 2.2.

Ø 1.13 The first reviewer's comments:Materials and Methods: “(…) 24-well plates coated with different materials (rat tail collagen, polylysine and Matrigel) (…)”. How this coating materials were selected? With what bases? One of the most used coating for fish muscle cell cultures is the combination of poly-lysine and laminin, which show high affinity for myoblast (and possibly could prevent the wash out seen at the immunofluorescence). I wonder why does the laminin was not selected in the study?

Response: Thank you for your constructive and helpful suggestion. In the early stage, our team conducted experiments on a variety of coated materials by consulting relevant data, including matrix glue, fish gelatin, pig gelatin, laminin, polylysine and rat tail collagen, etc. Through comparison, it was found that these three materials had a good affinity to muscle cells of other species, so we decided to conduct PSCs experiments on several materials with the best effect.

Ø 1.14 The first reviewer's comments:Materials and Methods: “Statistical significance of comparisons between the two groups was analyzed using multiple t-tests”. Which two groups? There are 3 different media (DMEM, F10 and DMEM/F12) and 3 different coating materials (collagen, polylysine and matrigel). Why did not used ANOVA (parametric) or Kruskall-Wallis (non-parametric)?

Response: Thank you for your constructive and helpful suggestion. The two groups in this article refer to pairwise comparisons. Comparison between basal media, namely DMEM and DMEM/F12 comparison, DMEM and F10 comparison, DMEM/F12 and F10 comparison; Comparison between coated materials, namely collagen and polylysine comparison, collagen and matrigel comparison, polylysine and matrigel comparison, so we used t-tests.

Ø 1.15 The first reviewer's comments:Results: The numbers of the figures indicated in the text do not correspond to the images in the manuscript. Review all of them.

Response: Thank you for your constructive and helpful suggestion. The numbers of the figures indicated in the text correspond to the images in the manuscript.

Ø 1.16 The first reviewer's comments:Results: The numbers of the figures indicated in the text do not correspond to the images in the manuscript. Review all of them.

Response: Thank you for your constructive and helpful suggestion. The sentence has been changed to “In conclusion, F10 or DMEM/F12 could significantly promote the long-term proliferation of PSCs by regulating cell cycle and enhancing cell proliferative activity, compared to DMEM. F10 has a more potent effect, compared to DMEM/F12”. Please see highlight in the “Results” section.

Ø 1.17 The first reviewer's comments:Results: In the Figure 1 (wrongly indicated as figure 2), the quality of the images could be better.

Response: Thank you for your constructive and helpful suggestion. The image has been optimized.

Ø 1.18 The first reviewer's comments:Results: In section 3.2, the end of the first paragraph and the beginning of the second paragraph are very repetitive. It is necessary to synthesize the information.

Response: Thank you for your constructive and helpful suggestion. In section 3.2, We explored the effects of different base media and different coated materials on cell proliferation from the total number of cells, population doubling time and expansion folds respectively. But the sentences in the text are not smooth, and the full text x is modified as follows:

“According to the above experimental results, F10 and DMEM/F12 can significantly promote the proliferation of PSCs under uncoated conditions. At present, most researchers have coated the culture vessels when conducting cell experiments. In this study, three commonly used coating materials (matrix, rat tail collagen and polylysine) were used to investigate the effects of different basal media on the proliferation of PSCs. Cells were cultured for 5 days. On the 5th day of culture, the morphology of PSCs cultured in F10 and DMEM/F12 basal medium under both coated and uncoated conditions showed a slender normal morphology (Fig. 3A), Both F10 and DMEM/F12 increased cell viability at day 5. Considering only the single factor of different coating conditions, in F10, the total number of PSCs cultured with the uncoated, polylysine-coated and matrix-coated  were 1.2, 1.1 and 1.4 times than that rat tail collagen-coated , respectively, and it showed that the effect of matrix-coated was the best; In DMEM/F12, the uncoated, polylysine-coated and matrix-coated cells were about 1.5 times, 1.3 times and 1.4 times than those of rat tail collagen-coated, respectively, and it showed that the effect of the uncoated was the best; Considering only the single factor of different basal media, F10 was about 1.5, 1.8 ,1.6 and 1.6times that of DMEM/F12 in uncoated, rat tail collagen-coated, polylysine-coated and matrix-coated repsectively. F10 was superior to DMEM/F12 under all conditions, among which matrix-coated was the best (Fig. 3B).

As shown in Fig. 3C, in F10, the total number of PSCs cultured with the uncoated increased by about 42 times, rat tail collagen-coated, polylysine-coated and matrix-coated was increased by about 19, 26 and 41 times, respectively; in DMEM/F12, uncoated, rat tail collagen-coated, polylysine-coated and matrix-coated was increased by about 21, 17, 19 and 24 times respectively. The proliferation of PSCs cultured in F10 basal medium and uncoated condition was the best, followed by that cultured in F10 basal medium and matrix-coated, and the two effects were similar. In addition, the least population doubling time was observed in PSCs cultured with F10 basal medium and matrix-coated at day 3 and cultured with F10 basal medium and uncoated or matrix-coated at day 5 (Fig. 3D). In conclusion, whether cultured in F10 or DMEM/ F12, matrix had the best proliferation effect on PSCs among the three coating materials used in this study, and the proliferation effect of uncoated and coated cells was similar.”

Ø 1.19 The first reviewer's comments:Results: “(…) and the fusion index reached about 20%, while that of DMEM/F12 was only about 10% (Fig. 4A and 4B)”. Explain in the "Methods" section how the fusion index was analyzed and obtained.

Response: Thank you for your constructive and helpful suggestion. Please see highlight in section 2.6. “Nuclei counts were determined by quantifying DAPI staining using ImageJ software. The fusion index was determined by quantifying nuclei within desmin-stained myofibres as a proportion of total nuclei and multiplying by 100 [11]”

Ø 1.20 The first reviewer's comments:“Alanyl-glutamine is very stable in aqueous solution and does not degrade spontaneously. This composition might be an underlying mechanism for its great support for cell propagation and colony-formation in the present study”. In addition, F10 and DMEM/F12 media could not show better results due to their lower concentration of glutamine compared to DMEM?

Ø 1.21 The first reviewer's comments:Discussion: The last paragraphs on page 10 are poorly written.

Ø 1.22 The first reviewer's comments:Discussion: “In this study, the concentrations of amino acids and vitamins in DMEM were much more abundant than in DMEM/F12 and F10”.

Considering the higher amount of amino acid and vitamins in DMEM, the authors should discuss better how this was the medium with the less promising results.

Ø 1.23 The first reviewer's comments:Discussion: This section needs to be further explored. How the achievements of the work will be important to meat production, largely cited in the introduction?

Response: Thank you for your constructive and helpful suggestion. The “Discussion” section has been rewritten as you suggested. Full text as below:

“In this study, we obtained a proliferation medium based on DMEM/F12 or F10 basal medium that effectively promoted the proliferation of muscle satellite cells from Larimichthys crocea in vitro for the first time. By analysis of expanded cell number, population doubling time, expression of cell proliferation marker genes cdk1 and cdk2 and myogenic differentiation potential, we demonstrated F10 and DMEM/F12 significantly increased cell viability, the total cell number of PSCs cultured with DMEM increased by about 22-fold, that cultured with F10 increased by about 31-fold, and that cultured with DMEM/F12 increased by about 27-fold, and the proliferation of PSCs in the growth medium based on DMEM/F12 or F10 did not affect their later differentiation, no matter whether the basic growth medium was DMEM/F12 or F10, more myotubes could be formed, the fusion index reached about 20%, while that of DMEM/F12 was about 10%.

Differentiation potential of stem cells decrease rapidly during long-term in vitro culture with increases in apoptosis and cell senescence[12-14]. Wagner et al have reported that mesenchymal stem cell differentiation potential declined and senescence-associated genes were increasingly expressed over time[15]. Qingzi lei et al also have found that the decline in differentiation potential of porcine muscle stem cells to be related to culture time rather than to cell division number before differentiation[16]. Therefore, it is essential to shorten cell population doubling time, producing as many cells as possible within a limited time. Such an approach would be very beneficial to industrialized production of cultured meat. The previous exploration of proliferation medium for PSCs, we found that when we chose DMEM for proliferation, when the cells grew to a certain number of generations, the cell viability and growth rate decreased. When the basic medium was replaced by F10 from DMEM, the cell viability and growth rate increased. It was found that different basal media had a great effect on the cells, so on this basis, two basal media were designed, namely F10 and DMEM/F12, to explore their effects on the growth of muscle satellite cells. In this study, the result showed that the total cell number of PSCs in F10 approximately 1.4-fold and DMEM/F12 approximately 1.6-fold than DMEM.

Different basal media have diverse functional effects. Related studies have shown that DMEM was used for cell proliferation, whereas, DMEM/F12 was used for cell differentiation and Ham's FI0 which contained nutrients such as vitamins, amino acids, and metabolites was suitable for long-term serum-free cultures [17]. Basal media vary in their composition and content, leading to their utilization in different culture conditions. In this study, the concentrations of amino acids and vitamins in DMEM were much more abundant than in DMEM/F12 and F10. Fayaz and Honaramooz [18] have showed that DMEM sufficiently supported in vitro propagation of gonocytes and somatic cells. Pahlavanneshan, et al. [19] have proved that Ham’s F10 medium provided better conditions for 2D culture of primary islet cells. Hua, et al. [20] have showed that among four different media (F12, DMEM/F12, 1640 and DMEM), DMEM/F12 medium was the most suitable media for lower-serum adherent culture. Wei, et al. [21] have indicated that glutamine was the key molecule to maintain cell growth and survival in culture and glutamine was over 2.5 mM in DMEM/F12 and DMEM/HG. In this study, DMEM and DMEM/F12 contained 4 and 2.5 mM glutamine, respectively. F10 contained 1.5mM alanyl-glutamine. This composition might be a key factor in promoting PSCs proliferation in our study. Singh, et al. [19] investigated the effects on cell growth by comparing different components of DMEM/F12 medium with DMEM medium. Hua, et al. [20] have showed that among four different media (F12, DMEM/F12, 1640 and DMEM), DMEM/F12 medium was the most suitable media for lower-serum adherent culture.

Cell adhesion and migration are essential for cell proliferation, some researchers have changed the structure of cell contact media to promote cell adhesion. Du, et al. [22] grafted silk sericin (SS) onto the surface of thermoplastic polyurethane (TPU) membrane by -NH2 bridge to build a high efficiency cell culture membrane which could promote cell adhesion. Dhania, et al. [23] have showed that different cells- Vero, Hela and MDBK cell lines cultured on the porous mats of fabricated polyhydroxyalkanoates blend scaffolds exhibited significant increase in cell viability and attachment over time. In this study, we adopted the latter approach by applying a coating material to the cell contact medium to promote cell attachment. We selected three common coating materials: rat tail collagen, Matrigel and poly-lysine. Among them, Matrigel is a common matrice and widely used to mimic the environment of the extracellular matrix (ECM) which assist in maintaining optimal balance for a series of adjusting cell behaviors including cell proliferation, cell migration, and cell differentiation [24]. The extracellular matrix (ECM) plays a major role in cell-cell and cell-matrix signaling during normal physiology and disease [25]. The results showed that there was no significant difference in the proliferation of PSCs between coated and uncoated materials, and the effect of matrix glue was the best among the coated materials. The reason for this phenomenon may be related to species or cell types, indicating that the nutritional components in the proliferation medium based on these three basic media can meet the growth of PSCs. There is no need to provide it with additional external adhesion or migration forces.”

Reviewer 2 Report

This study aimed to explore the effects of different basal media and coating materials on the proliferation and differentiation of PSCs. The results demonstrated that F10 and DMEM/F12 were superior to DMEM high sugar, showing enhanced cell proliferation rates and normal cell morphology. The data presented showed that PSCs maintained their stemness and were able to differentiate into more myotubes when grown in DMEM/F12 or F10-based growth media. Finally, the authors demonstrated that matrix coating was the most effective in promoting cell proliferation and facilitating the formation of more myotubes. The manuscript is well-written, and the data supports the conclusion. Hence, I endorse the publication of this manuscript in its current form.  

Moderate English editing is required. 

Author Response

Second reviewer's comments: This study aimed to explore the effects of different basal media and coating materials on the proliferation and differentiation of PSCs. The results demonstrated that F10 and DMEM/F12 were superior to DMEM high sugar, showing enhanced cell proliferation rates and normal cell morphology. The data presented showed that PSCs maintained their stemness and were able to differentiate into more myotubes when grown in DMEM/F12 or F10-based growth media. Finally, the authors demonstrated that matrix coating was the most effective in promoting cell proliferation and facilitating the formation of more myotubes. The manuscript is well-written, and the data supports the conclusion. Hence, I endorse the publication of this manuscript in its current form.

Response: Thank you for your recognition of my research content and the writing of the article. There may be some problems in the article that need to be modified, and I will revise it carefully to meet the quality requirements of publication. Thank you.

Reviewer 3 Report

I thank the authors for the work entitles “Characterization of Proliferation Medium and its Effect on Differentiation of Muscle Satellite Cells from Larimichthys crocea”. Below my comments:

Title: 

Title has to indicate that the final and future application of the study is cell cultured fish production

Introduction:

- Please explain and rephrase this sentence (What other components? Which same kind of cells?): “Selection of appropriate culture medium is crucial for cell culture in vitro. Different basal media are used in different culture conditions because of their different components and content. So far, there is no relevant report on the study of different basal media on the proliferation of the same kind of cells. In the case of the same other components of the medium, choosing different basic media, the results of the effects on proliferation rate, cell morphology and stemness of PSCs were quite different”

- If there is, please report more work on cultured fish production other than tuna.

Material and Methods:

- Chapter 2.5: Title should report both, cell proliferation and differentiation. Also, please mention the concentration of the different coated materials. 

- Chapter 2.6: Please describe the seeding of the cells. 

Results:

- Figure 1 not present? It appears that figures numbers do not match the citation of the same figures in the text.

- Chapter 3.1, Figure 2 and Figure 3: Please explain how did you count the cells. Was it a manual counting or an evaluation of the area covered by the cells of the electron microscopic images? (Images do not show striking differences). Please, describe and comment especially the expansion fold and population doubling time analysis.

- Chapter 3.2: In contrast to the title of the chapter, obtained results indicate that coated materials do not show any advantage on proliferation in comparison to uncoated cells. Please comment.

- Chapter 3.3: Please explain the sentence: “During the proliferation process, the cells cultured in DMEM basal medium did not grow”, when results in Figure 2 do not show that, at least not so evidently. 

- Please explain the “fusion index” and comment the low percentage generally obtained (10% and 20%)

Discussion:

- It is very difficult to read, especially the comparison of the different media. Please, substitute that part with a table.

“At present, most research focuses on the screening of cytokines or growth factors in culture medium.”Please shortly comment their results.

VLCAD-deficient cell lines: No description is given in the text

Conclusions:

- The main argument is about the coated material Matrigel which appears to be favorable during the staining (no wash out). Is it possible that the staining was performed in too hard conditions? 

After all, cells appear to differentiate and to survive medium change and wash for 3 days, using all coated materials.

Author Response

Responses to comments of Editor

Thank you for your serious and constructive comments on our manuscript. According to your suggestion, the manuscript has been revised as a letter to editor. The revisions we have made are as follows:

Ø 3.1 The third reviewer's comments:

Title: Title has to indicate that the final and future application of the study is cell cultured fish production.

Response: Thank you for your constructive and helpful suggestion. The title has been changed to “Characterization of Proliferation Medium and its Effect on Differentiation of Muscle Satellite Cells from Larimichthys crocea in Cultured Fish Meat Production”,

Ø 3.2 The third reviewer's comments:Introduction:

Please explain and rephrase this sentence (What other components? Which same kind of cells?): “Selection of appropriate culture medium is crucial for cell culture in vitro. Different basal media are used in different culture conditions because of their different components and content. So far, there is no relevant report on the study of different basal media on the proliferation of the same kind of cells. In the case of the same other components of the medium, choosing different basic media, the results of the effects on proliferation rate, cell morphology and stemness of PSCs were quite different”- If there is, please report more work on cultured fish production other than tuna.

Response: Thank you for your constructive and helpful suggestion. Cell types include most cells, and these media are generic for these cells, so the specific cell type is not indicated here. The sentence has been changed to “Selection of appropriate culture medium is crucial for cell culture in vitro. Basal media vary in their composition and content, leading to their utilization in different culture conditions.

 Such as, compared with DMEM medium, DMEM/F12 contained more nutrients, mainly in amino acids including L-alanine, L-aspartate and L-cysteine;The vitamins contain biotin and vitamin B12; The inorganic salts contain CuSO4·5H2O and ZnSO4·7H2O; It also contains thymidine, hypoxanthine sodium and linoleic acid.” Please see the highlight in the “Introduction” section.

Ø 3.3 The third reviewer's comments:Material and Methods:

Chapter 2.5: Title should report both, cell proliferation and differentiation. Also, please mention the concentration of the different coated materials.

Response: Thank you for your constructive and helpful suggestion

The title has been changed to “Cell proliferation and differentiation”, and the concentration of the different coated materials has been added (rat tail collagen 12μg/mL, polylysine 0.1 mg/mL and Matrigel 1 mg/mL). Please see the highlighted in section 2.5.

Ø 3.4 The third reviewer's comments:- Chapter 2.6: Please describe the seeding of the cells.

Response: Thank you for your constructive and helpful suggestion. This section mainly introduces the methods of immunofluorescence staining, and the specific methods of cell seeding are provided in Section 2.5 (At first, about 5×104 cells/well were prepared into cell suspension with different basal media (DMEM, F10 and DMEM/F12), and then seeded into 24-well plates coated with different materials (rat tail collagen 12μg/mL, polylysine 0.1 mg/mL and Matrigel 1 mg/mL) for proliferation and culture for 3 days).

Ø 3.5 The third reviewer's comments:Results:

- Figure 1 not present? It appears that figures numbers do not match the citation of the same figures in the text.

Response: Thank you for your constructive and helpful suggestion

We have modified, the numbers of the figures indicated in the text have been correspond to the images in the manuscript.

Ø 3.6 The third reviewer's comments:- Chapter 3.1, Figure 2 and Figure 3: Please explain how did you count the cells. Was it a manual counting or an evaluation of the area covered by the cells of the electron microscopic images? (Images do not show striking differences). Please, describe and comment especially the expansion fold and population doubling time analysis.

Response: Thank you for your constructive and helpful suggestion. The number of cells is converted by absorbance (please see the section 2.3 CCK-8 assay). Expansion fold refers to the ratio of the measured time point to the number of cells on the first day; population doubling time refers to the ratio of the measured time point to the previous measured time point; Both of them were parameters reflecting cell viability, they all described in the result.

Ø 3.7 The third reviewer's comments:- Chapter 3.2: In contrast to the title of the chapter, obtained results indicate that coated materials do not show any advantage on proliferation in comparison to uncoated cells. Please comment.

Response: Thank you for your constructive and helpful suggestion. The reason for this phenomenon may be related to species or cell types, indicating that the nutritional components in the proliferation medium based on these three basic media can meet the growth of PSCs. There is no need to provide it with additional external adhesion or migration forces.

The title has some inducement, the title has been changed to “Matrigel coating among the three coated material can promote the proliferation of PSCs”.

Ø 3.8 The third reviewer's comments:- Chapter 3.3: Please explain the sentence: “During the proliferation process, the cells cultured in DMEM basal medium did not grow”, when results in Figure 2 do not show that, at least not so evidently.

Response: Thank you for your constructive and helpful suggestion. DMEM/F12 basal medium or F10 basal medium had a better effect on cell proliferation when we discussed the influence of different basal medium on cell proliferation in the early stage, so we always used DMEM/F12 basal medium for cell culture. It may be due to the domestication of cells, the cells did not grow after being changed to DMEM medium. the cell growth basal medium image was showed as below

I hope my answer has solved your doubts and made you satisfied. Thank you.

Ø 3.9 The third reviewer's comments:- Please explain the “fusion index” and comment the low percentage generally obtained (10% and 20%)

Response: Thank you for your constructive and helpful suggestion. The fusion index was determined by quantifying nuclei within desmin-stained myofibres as a proportion of total nuclei and multiplying by 100. The low fusion index may be related to species or differentiation medium formulation. Because the growth environment of Larimichthys crocea is different from that of our common mammals, its differentiation efficiency is not the same, and the general differentiation medium used in Larimichthys crocea differentiation efficiency is very low.

Ø 3.10 The third reviewer's comments:Discussion:

- It is very difficult to read, especially the comparison of the different media. Please, substitute that part with a table.- “At present, most research focuses on the screening of cytokines or growth factors in culture medium.”Please shortly comment their results.- VLCAD-deficient cell lines: No description is given in the text

Response: Thank you for your constructive and helpful suggestion. The “Discussion” section has been rewritten as you suggested. Please see the section 4 “Discussion” of the manuscript. The “Discussion” section has been rewritten as you suggested. This section is briefly described as below:

The first paragraph discussed the results from cell viability, proliferation and differentiation; The second paragraph mainly explained that the longer the time of cell culture in vitro, the cell differentiation ability and vitality will decline, so it is necessary to obtain a high-efficiency proliferation medium; Which leads to the third paragraph, the third paragraph is about exploring an efficient proliferation medium from different basal medium; In order to further improve the cell proliferation efficiency, consider the use of coated materials, the fourth paragraph describes the coating material. Full text as below:

“In this study, we obtained a proliferation medium based on DMEM/F12 or F10 basal medium that effectively promoted the proliferation of muscle satellite cells from Larimichthys crocea in vitro for the first time. By analysis of expanded cell number, population doubling time, expression of cell proliferation marker genes cdk1 and cdk2 and myogenic differentiation potential, we demonstrated F10 and DMEM/F12 significantly increased cell viability, the total cell number of PSCs cultured with DMEM increased by about 22-fold, that cultured with F10 increased by about 31-fold, and that cultured with DMEM/F12 increased by about 27-fold, and the proliferation of PSCs in the growth medium based on DMEM/F12 or F10 did not affect their later differentiation, no matter whether the basic growth medium was DMEM/F12 or F10, more myotubes could be formed, the fusion index reached about 20%, while that of DMEM/F12 was about 10%.

Differentiation potential of stem cells decrease rapidly during long-term in vitro culture with increases in apoptosis and cell senescence[12-14]. Wagner et al have reported that mesenchymal stem cell differentiation potential declined and senescence-associated genes were increasingly expressed over time[15]. Qingzi lei et al also have found that the decline in differentiation potential of porcine muscle stem cells to be related to culture time rather than to cell division number before differentiation[16]. Therefore, it is essential to shorten cell population doubling time, producing as many cells as possible within a limited time. Such an approach would be very beneficial to industrialized production of cultured meat. The previous exploration of proliferation medium for PSCs, we found that when we chose DMEM for proliferation, when the cells grew to a certain number of generations, the cell viability and growth rate decreased. When the basic medium was replaced by F10 from DMEM, the cell viability and growth rate increased. It was found that different basal media had a great effect on the cells, so on this basis, two basal media were designed, namely F10 and DMEM/F12, to explore their effects on the growth of muscle satellite cells. In this study, the result showed that the total cell number of PSCs in F10 approximately 1.4-fold and DMEM/F12 approximately 1.6-fold than DMEM.

Different basal media have diverse functional effects. Related studies have shown that DMEM was used for cell proliferation, whereas, DMEM/F12 was used for cell differentiation and Ham's FI0 which contained nutrients such as vitamins, amino acids, and metabolites was suitable for long-term serum-free cultures [17]. Basal media vary in their composition and content, leading to their utilization in different culture conditions. In this study, the concentrations of amino acids and vitamins in DMEM were much more abundant than in DMEM/F12 and F10. Fayaz and Honaramooz [18] have showed that DMEM sufficiently supported in vitro propagation of gonocytes and somatic cells. Pahlavanneshan, et al. [19] have proved that Ham’s F10 medium provided better conditions for 2D culture of primary islet cells. Hua, et al. [20] have showed that among four different media (F12, DMEM/F12, 1640 and DMEM), DMEM/F12 medium was the most suitable media for lower-serum adherent culture. Wei, et al. [21] have indicated that glutamine was the key molecule to maintain cell growth and survival in culture and glutamine was over 2.5 mM in DMEM/F12 and DMEM/HG. In this study, DMEM and DMEM/F12 contained 4 and 2.5 mM glutamine, respectively. F10 contained 1.5mM alanyl-glutamine. This composition might be a key factor in promoting PSCs proliferation in our study. Singh, et al. [19] investigated the effects on cell growth by comparing different components of DMEM/F12 medium with DMEM medium. Hua, et al. [20] have showed that among four different media (F12, DMEM/F12, 1640 and DMEM), DMEM/F12 medium was the most suitable media for lower-serum adherent culture.

Cell adhesion and migration are essential for cell proliferation, some researchers have changed the structure of cell contact media to promote cell adhesion. Du, et al. [22] grafted silk sericin (SS) onto the surface of thermoplastic polyurethane (TPU) membrane by -NH2 bridge to build a high efficiency cell culture membrane which could promote cell adhesion. Dhania, et al. [23] have showed that different cells- Vero, Hela and MDBK cell lines cultured on the porous mats of fabricated polyhydroxyalkanoates blend scaffolds exhibited significant increase in cell viability and attachment over time. In this study, we adopted the latter approach by applying a coating material to the cell contact medium to promote cell attachment. We selected three common coating materials: rat tail collagen, Matrigel and poly-lysine. Among them, Matrigel is a common matrice and widely used to mimic the environment of the extracellular matrix (ECM) which assist in maintaining optimal balance for a series of adjusting cell behaviors including cell proliferation, cell migration, and cell differentiation [24]. The extracellular matrix (ECM) plays a major role in cell-cell and cell-matrix signaling during normal physiology and disease [25]. The results showed that there was no significant difference in the proliferation of PSCs between coated and uncoated materials, and the effect of matrix glue was the best among the coated materials. The reason for this phenomenon may be related to species or cell types, indicating that the nutritional components in the proliferation medium based on these three basic media can meet the growth of PSCs. There is no need to provide it with additional external adhesion or migration forces.”

Ø 3.11 The third reviewer's comments:Conclusions:

- The main argument is about the coated material Matrigel which appears to be favorable during the staining (no wash out). Is it possible that the staining was performed in too hard conditions?  

After all, cells appear to differentiate and to survive medium change and wash for 3 days, using all coated materials.

Response: Thank you for your constructive and helpful suggestion. Myotubes are very easy to flush out, so we're very careful during the staining process, Moreover, it is used in the dyeing process to fix the polymethanol. Therefore, the main reason may still be related to the strong adhesion of matrix to cells, of course, we will further explore this issue in the future.

Round 2

Reviewer 1 Report

The manuscript has been substantially improved. Most of the points raised were answered satisfactorily.

I still think it is very strange that the authors did not cite references related to fish muscle cell cultures or even fish muscle, especially because they are submiting the manuscript to a journal named "Fishes".
But I understood that the authors focused mainly on the culture technique, needed for the in vitro meat production, as justified.

I believe this issue can be more adequately evaluated by the editorial board.

Author Response

The first reviewer's comments:I still think it is very strange that the authors did not cite references related to fish muscle cell cultures or even fish muscle, especially because they are submiting the manuscript to a journal named "Fishes".But I understood that the authors focused mainly on the culture technique, needed for the in vitro meat production, as justified.

Response: Thank you for your constructive and helpful suggestion

According to your suggestion, we have made corresponding modifications. In the second paragraph of the “Introduction” section, we have introduced the content related to fish muscle. The content as below:

“Most of the muscles of L. crocea belong to the white muscle of vertebrates. It is mainly composed of skeletal muscle and fat[3]. Fish skeletal muscle cell lines are often described simply by their predominate cell shape, which usually has been described as either fibroblast-like, epithelial-like or spindle-like[4]. All these cell lines can contribute to studies on in vitro meat production and, both their ante factum properties, such as the species and anatomical site from which the cell line arose, and post factum properties, such as growth factor requirements, are important features to consider[5]. Muscle satellite cells belong to fibroblast-like and have exhibited the greatest potential which can differentiate into skeletal muscle cells[6]. They are generally quiescent until to be activated and become myoblasts, which will undergo proliferation and myogenic differentiation to form new muscle fibers[3,7]. Cultured meat is an efficient, safe and sustainable meat production technology[8]. In 2013, the first beef burger made from cell-cultured meat was introduced[9]. In 2019, China's first cultured pork meat was produced by Professor Zhou Guanghong. Memphis meat (later known as UPSIDE Foods) made meatballs using cultured beef, chicken, and duck[2,10]. Finless Foods successfully manufactured cultured meat patties using tuna stem cells. Also, with the rise and maturity of 3D technology, several researchers have successfully constructed meat tissues of livestock such as cow[11] and pig[12] using it. However, unlike livestock, few studies have used marine fish to explore in vitro myogenesis[3].”

In addition, in order to make the article more rigorous, through our repeated checks, we have revised the “Abstract” section and some “Title”. The content as below:

(1) Abstract: To find a suitable medium for muscle satellite cells of Larimichthys crocea, herein, the effect of different basal media and coating materials on the proliferation of the piscine satellite cells (PSCs) were explored. Firstly, two basal media namely F10 and DMEM/F12 were selected as experimental materials and DMEM high sugar was the main culture medium used for fish muscle cell cultures as a control, the results showed that the cells proliferated better in F10 and DMEM/F12 than in DMEM high sugar, and F10 has a more potent effect, with normal cell morphology and high proliferation rate. On this basis, we also investigated the effects of rat tail collagen-coated, polylysine-coated and matrix-coated compared with uncoated on the proliferation and differentiation of PSCs. The results indicated that there was no significant difference between substrate coated and uncoated on the proliferation of PSCs based in F10 medium, and it was found that the myotubes were washed out, and only Matrigel-coated was intact in the process of differentiation. This results also showed that PSCs could still differentiate into myotubes without affecting their stemness after proliferation in F10-based growth medium. This study obtained an efficient proliferation medium based on F10. It could shorten the culture time and maintain the stemness of PSCs, providing a basis for large-scale cell expansion and cell culture meat production in the future.

(2) Title: 3.2. F10 basal medium significantly promoted PSCs proliferation

3.3. Effect of coated and uncoated on the proliferation of PSCs

3.4. F10 didn’t affect the stemness of PSCs

I hope my modification has solved your doubts and made you satisfied. Thank you.

Reviewer 3 Report

I thank the authors for carefully answering all the points raised in the previous revision. Although the paper has highly improved, I still would like to present the following points:

  1. While I understand the effect of different media on PSCs, unfortunately, results obtained using the coated materials are not sufficient and robust enough. Please comment 
  2. The end of the introduction needs to report a short sentence with a summary of your results and the advantage of your research against the state of the art. Beside this, as I mentioned in my first review, please report any other work on in vitro fish meet production from PSCs, if relevant. 
  3. The data shown in figure 1 is not explained in the results section
  4. The discussion has greatly improved. However, now it includes many descriptive parts rather than an analysis of the results. Please revise

Please check the form and structure of the newly added sentences especially in the conclusion part

Author Response

3.1 The third reviewer's comments: While I understand the effect of different media on PSCs, unfortunately, results obtained using the coated materials are not sufficient and robust enough. Please comment

Response: Thank you for your constructive and helpful suggestion

There may have been some problems in the description and logic of the results before, which caused the deviation in your understanding. For this, we have made corresponding modifications according to your suggestions. In the section of “3.3 Effect of coated and uncoated on the proliferation of PSCs (We have also modified the title, and this is the revised title)” with highlights, we have made additions and modifications. The content as below:

“According to the above experimental results, both F10 and DMEM/F12 can significantly promote PSCs proliferation under uncoated conditions. At present, most researchers have coated the culture vessels when conducting cell experiments. Herein, three commonly used coating materials (matrix, rat tail collagen and polylysine) were used to investigate the effects of different basal media on the proliferation of PSCs. Cells were cultured for 5 days. On the 5th day of culture, the morphology of PSCs cultured in F10 and DMEM/F12 basal medium under both coated and uncoated conditions showed a slender normal morphology (Fig. 3A). With the increase of culture days, the cell viability of DMEM/F12 and F10 all increased significantly (Fig. 3B). As shown in Fig. 3C, for calculating expansion fold, the total number of cells on day 5 was compared with the total number of cells on day 1. In F10, PSCs cultured with the uncoated, rat tail collagen-coated, polylysine-coated and matrix-coated were increased by about 42, 19, 26 and 41 times respectively, this result showed that there was no significant difference in uncoated group and matrix-coated group(P>0.05). Whereas, in DMEM/F12, the expansion fold in uncoated, rat tail collagen-coated, polylysine-coated and matrix-coated group were increased by about 21, 17, 19 and 24 times respectively. this result showed that matrix-coated was the best, uncoated as follow, which have significant difference (P<0.001). The reason for the result may be due to the different composition between DMEM/F12 and F10.

In addition, as shown in Fig. 3D, the population doubling time of day 5 compared with day 1. In F10, PSCs cultured with the uncoated, rat tail collagen-coated, polylysine-coated and matrix-coated required about 18, 23, 20 and 18 hours respectively, this result showed that both uncoated and matrix-coated required the least population doubling time (P>0.05). Whereas, in DMEM/F12, PSCs cultured with the uncoated, rat tail collagen-coated, polylysine-coated and matrix-coated required about 22, 24, 23 and 21 hours respectively, this result showed that uncoated was the best, matrix-coated as follow, and there was no significant difference in their effectiveness(P>0.05).

In conclusion, whether cultured in F10 or DMEM/ F12, matrix had the best proliferation effect on PSCs among the three coating materials used in this study, Additionally, F10 medium is more conducive to cell proliferation. Under these conditions, there was no significant difference between substrate coated and uncoated. However, in DMEM/F12 medium, there was a slight difference in cell proliferation between matrix coated and uncoated, suggesting that differences in medium composition can give rise to differences in cell proliferation on the coated material.”

I hope my modification has solved your doubts and made you satisfied. Thank you.

Ø 3.2 The third reviewer's comments: The end of the introduction needs to report a short sentence with a summary of your results and the advantage of your research against the state of the art. Beside this, as I mentioned in my first review, please report any other work on in vitro fish meet production from PSCs, if relevant.

Response: Thank you for your constructive and helpful suggestion

According to your suggestion, we have made corresponding modifications. In the end of the “Introduction” section with highlights, we have made additions and modifications. The content as below:

“Shortening the culture time is very important for stem cell culture in vitro, because differentiation potential of stem cells decrease rapidly during long-term in vitro culture with increases in apoptosis and cell senescence[16-18]. Meanwhile, obtaining numerous of stem cells is a pivotal step for cultured meat. It is essential to obtained an efficient proliferation medium. However, at present, there are few reports on the optimization of proliferation medium, and most researchers have focused on suspension culture depended on microcarrier[19,20] for large-scale expansion. Although these methods could obtain enough cells, the culture cycle was still very long. In this study, we have obtained an excellent medium based on F10 basal medium significantly shorten the culture time of PSCs from L. crocea in vitro compared with the traditional culture medium DMEM and maintain it stemness. It can provide enough seed cells for the subsequent production of cultured meat.”

Also, in the second paragraph of the “Introduction” section, we have supplemented the content related some work on in vitro fish meet production from PSCs. The content as below:

“Most of the muscles of L. crocea belong to the white muscle of vertebrates. It is mainly composed of skeletal muscle and fat[3]. Fish skeletal muscle cell lines are often described simply by their predominate cell shape, which usually has been described as either fibroblast-like, epithelial-like or spindle-like[4]. All these cell lines can contribute to studies on in vitro meat production and, both their ante factum properties, such as the species and anatomical site from which the cell line arose, and post factum properties, such as growth factor requirements, are important features to consider[5]. Muscle satellite cells belong to fibroblast-like and have exhibited the greatest potential which can differentiate into skeletal muscle cells[6]. They are generally quiescent until to be activated and become myoblasts, which will undergo proliferation and myogenic differentiation to form new muscle fibers[3,7]. Cultured meat is an efficient, safe and sustainable meat production technology[8]. In 2013, the first beef burger made from cell-cultured meat was introduced[9]. In 2019, China's first cultured pork meat was produced by Professor Zhou Guanghong. Memphis meat (later known as UPSIDE Foods) made meatballs using cultured beef, chicken, and duck[2,10]. Finless Foods successfully manufactured cultured meat patties using tuna stem cells. Also, with the rise and maturity of 3D technology, several researchers have successfully constructed meat tissues of livestock such as cow[11] and pig[12] using it. However, unlike livestock, few studies have used marine fish to explore in vitro myogenesis[3].”

I hope my modification has solved your doubts and made you satisfied. Thank you.

Ø 3.3 The third reviewer's comments: The data shown in figure 1 is not explained in the results section

Response: Thank you for your constructive and helpful suggestion

According to your suggestion, we have added the section “3.1 Isolation and identification of PSCs”. The content as below:

“The isolated cells were characterized by immunofluorescent staining with the specific myoblast markers such as Pax7 and MyoD1. The results showed that about 99% of the cells were Pax7 positive, and 99% of the cells were MyoD1 positive (Fig. 1). These data suggested that the isolated cells possessed the properties of PSCs.”

I hope my modification has solved your doubts and made you satisfied. Thank you.

Ø 3.4 The third reviewer's comments: The discussion has greatly improved. However, now it includes many descriptive parts rather than an analysis of the results. Please revise

Response: Thank you for your constructive and helpful suggestion

According to your suggestion, we have made corresponding modifications. In the end of the “Discussion” section with highlight, we have made additions and modifications. The content as below:

“It is essential to obtain an efficient proliferation medium to shorten cell population doubling time, producing as many cells as possible within a limited time. Such an approach would be very beneficial to industrialized production of cultured meat. As some researchers have reported that differentiation potential decrease rapidly, apoptosis and cell senescence increase during long-term in vitro culture, such as, Wagner et al have reported that mesenchymal stem cell differentiation potential declined and senescence-associated genes were increasingly expressed over time[22]. Qingzi lei et al also have found that the decline in differentiation potential of porcine muscle stem cells to be related to culture time rather than to cell division number before differentiation[23]. The previous exploration of proliferation medium for PSCs, we found that when we chose DMEM for proliferation, when the cells grew to a certain number of generations, the cell viability and growth rate decreased. When the basic medium was replaced by F10 from DMEM, the cell viability and growth rate increased. It was found that different basal media had a great effect on the cells, so on this basis, two basal media were designed, namely F10 and DMEM/F12, to explore their effects on the growth of muscle satellite cells. In this study, we obtained an efficient proliferation medium based on DMEM/F12 or F10 basal medium that effectively promoted the proliferation of muscle satellite cells from L. crocea in vitro for the first time. By analysis of expanded cell number, population doubling time, expression of cell proliferation marker genes cdk1 and cdk2 and myogenic differentiation potential, we demonstrated F10 and DMEM/F12 significantly increased cell viability, the total cell number of PSCs cultured with DMEM, F10 and DMEM/F12 increased by about 22-fold, 31-fold and 27-fold respectively, and the proliferation of PSCs in the growth medium based on F10 did not affect their later differentiation, there are more myotubes could be formed, the fusion index reached about 20%.

Different basal media have diverse functional effects. Related studies have shown that DMEM was used for cell proliferation, whereas, DMEM/F12 was used for cell differentiation and Ham's FI0 which contained nutrients such as vitamins, amino acids, and metabolites was suitable for long-term serum-free cultures [24]. Basal media vary in their composition and content, leading to their utilization in different culture conditions. In this study, the concentrations of amino acids and vitamins in DMEM were much more abundant than in DMEM/F12 and F10. Fayaz and Honaramooz [25] have showed that DMEM sufficiently supported in vitro propagation of gonocytes and somatic cells. Pahlavanneshan, et al. [26] have proved that Ham’s F10 medium provided better conditions for 2D culture of primary islet cells. Hua, et al. [27] have showed that among four different media (F12, DMEM/F12, 1640 and DMEM), DMEM/F12 medium was the most suitable media for lower-serum adherent culture. Wei, et al. [28] have indicated that glutamine was the key molecule to maintain cell growth and survival in culture and glutamine was over 2.5 mM in DMEM/F12 and DMEM/HG. In this study, DMEM and DMEM/F12 contained 4 and 2.5 mM glutamine, respectively. F10 contained 1.5mM alanyl-glutamine. This composition might be a key factor in promoting PSCs proliferation in our study. Singh, et al. [19] investigated the effects on cell growth by comparing different components of DMEM/F12 medium with DMEM medium. Hua, et al. [20] have showed that among four different media (F12, DMEM/F12, 1640 and DMEM), DMEM/F12 medium was the most suitable media for lower-serum adherent culture.

Cell adhesion and migration are essential for cell proliferation, some researchers have changed the structure of cell contact media to promote cell adhesion. Du, et al. [29] grafted silk sericin onto the surface of thermoplastic polyurethane (TPU) membrane by -NH2 bridge to build a high efficiency cell culture membrane which could promote cell adhesion. Dhania, et al. [30] have showed that different cells- Vero, Hela and MDBK cell lines cultured on the porous mats of fabricated polyhydroxyalkanoates blend scaffolds exhibited significant increase in cell viability and attachment over time. In this study, we adopted the latter approach by applying a coating material to the cell contact medium to promote cell attachment. We selected three common coating materials: rat tail collagen, Matrigel and poly-lysine. Among them, Matrigel is a common matrice and widely used to mimic the environment of the extracellular matrix (ECM) which assist in maintaining optimal balance for a series of adjusting cell behaviors including cell proliferation, cell migration, and cell differentiation [31]. The ECM plays a major role in cell-cell and cell-matrix signaling during normal physiology and disease [32]. The results showed that there was no significant difference in the proliferation of PSCs between coated and uncoated materials, and the effect of matrix glue was the best among the coated materials. The reason for this phenomenon may be related to species or cell types, indicating that the nutritional components in the proliferation medium based on these three basic media can meet the growth of PSCs. There is no need to provide it with additional external adhesion or migration forces.”

I hope my modification has solved your doubts and made you satisfied. Thank you.

Ø 3.5 The third reviewer's comments: Please check the form and structure of the newly added sentences especially in the conclusion part

Response: Thank you for your constructive and helpful suggestion

According to your suggestions, we have checked and revised the English writing of the “Conclusion” section with highlight as well as the full text. The modifications of the “Conclusion” section as below:

“As we all known, obtaining enough stem cells in vitro is a pivotal step for cultured meat. For this purpose, this study compared with the effects of different basal media on the proliferation and differentiation of PSCs and finally obtained an efficient proliferation medium based on F10. The total cell number of PSCs cultured with F10 increased by about 31-fold, significantly higher than the traditional culture medium DMEM (22-fold). In order to promote cell adhesion and migration, we selected three common coating materials: rat tail collagen, Matrigel and poly-lysine. The result showed that there was no significant difference between substrate coated and uncoated on the proliferation of PSCs based in F10 medium, and it was found that the myotubes were washed out, and only the ones coated with Matrigel were intact in the process of differentiation, which may be caused by the strong adhesion ability of Matrigel. In the matrix coated condition, the fusion index of F10 reached about 20%. The proliferation of PSCs in the growth medium based on F10 did not affect its later differentiation. Taken together, in this study, we obtained an excellent proliferation medium, which could make the rapid proliferation and maintain the stemness of PSCs. This study provides a basis for large-scale cell expansion and cell culture meat production in the future.”

In addition, in order to make the article more rigorous, through our repeated checks, we have revised the “Abstract” section and some “Title”. The content as below:

(1) Abstract: To find a suitable medium for muscle satellite cells of Larimichthys crocea, herein, the effect of different basal media and coating materials on the proliferation of the piscine satellite cells (PSCs) were explored. Firstly, two basal media namely F10 and DMEM/F12 were selected as experimental materials and DMEM high sugar was the main culture medium used for fish muscle cell cultures as a control, the results showed that the cells proliferated better in F10 and DMEM/F12 than in DMEM high sugar, and F10 has a more potent effect, with normal cell morphology and high proliferation rate. On this basis, we also investigated the effects of rat tail collagen-coated, polylysine-coated and matrix-coated compared with uncoated on the proliferation and differentiation of PSCs. The results indicated that there was no significant difference between substrate coated and uncoated on the proliferation of PSCs based in F10 medium, and it was found that the myotubes were washed out, and only Matrigel-coated was intact in the process of differentiation. This results also showed that PSCs could still differentiate into myotubes without affecting their stemness after proliferation in F10-based growth medium. This study obtained an efficient proliferation medium based on F10. It could shorten the culture time and maintain the stemness of PSCs, providing a basis for large-scale cell expansion and cell culture meat production in the future.

(2) Title: 3.2. F10 basal medium significantly promoted PSCs proliferation

3.3. Effect of coated and uncoated on the proliferation of PSCs

3.4. F10 didn’t affect the stemness of PSCs

I hope my modification has solved your doubts and made you satisfied. Thank you.

Round 3

Reviewer 3 Report

I appreciate the modification of the title and a more objective analysis of the experiments involving the coated materials. 

However, the newly added sentences require a much better proofreading and editing.

Please, revise especially the sentences comparing the results between different media and their statistical significance (doubling time, marker gene expression, differentiation)

Newly modified sections (3.3, discussion, conclusion) contain many English mistakes, very long sentences and excessive (sometimes wring) use of punctuation. Please, rather use conjunctions/connectors.

All this makes the new added sentences very difficult to read and understand. Please proof-read them.

Author Response

Responses to comments of Editor

Thank you for your serious and constructive comments on our manuscript. According to your suggestion, the manuscript has been revised as a letter to editor. The revisions we have made are as follows:

Ø 3.1 The third reviewer's comments:I appreciate the modification of the title and a more objective analysis of the experiments involving the coated materials. However, the newly added sentences require a much better proofreading and editing. Please, revise especially the sentences comparing the results between different media and their statistical significance (doubling time, marker gene expression, differentiation)

Response: Thank you for your constructive and helpful suggestion

According to your suggestion, we have proofreading and editing the whole article especially the newly added sentences, and the part of modifications have been highlighted. The content of section 3.2 as below:

  • 2. F10 basal medium significantly promoted PSCs proliferation

To evaluate the effect of different basal media on the proliferation of PSCs, the cells were cultured for 18 days in uncoated 96-well plates. Meanwhile, the cell viability was determined by CCK-8 kit at different time points. The proliferation medium consisted of different basal media (DMEM, DMEM/F12 and F10), 10% FBS, 10 ng/ml bFGF and 1% PSA. As shown in Fig. 2A, in the early stage of proliferation (1-7 d), the morphology of cells cultured in DMEM/F12 and F10 basal medium was thin and long, while the morphology of cells cultured in DMEM was thick and short. Then, in the late stage of proliferation (after 7 d), they were all thin and long. This result showed that different basal media could maybe affect the development of PSCs.

With the increase of culture days, the cell density of F10 and DMEM/F12 was significantly higher than DMEM (Fig. 2B). As shown in Fig. 2C, the total number of cells on day 18 was compared with the total number of cells on day 1 to calculate expansion fold. The result showed that PSCs cultured with DMEM, F10 and DMEM/F12 increased by about 22-fold, 31-fold and 27-fold respectively. F10 had the best proliferation effect, followed by DMEM/F12 and DMEM. In addition, the population doubling time was calculated according to the ratio of cell viability on day 18 to day 1. The result showed that PSCs cultured with F10 showed the least population doubling time, followed by DMEM/F12 and DMEM (Fig. 2D). Similarly, compared with DMEM high sugar basal medium, the expression of cell cycle regulators Cdk1 and Cdk2 was increased about 10-fold when PSCs were cultured with F10 (Fig. 2E). Also, by comparing the ratio of G0/G1 to S+G2 in F10 and DMEM/F12, the value of F10 was less than DMEM/F12. This result demonstrated that F10 was more effective than DMEM/F12 in promoting cell proliferation (Fig. 2F).

In conclusion, F10 or DMEM/F12 could significantly promote the long-term proliferation of PSCs compared to DMEM. Similarly, F10 has a more potent effect compared to DMEM/F12.”

The above is just to show the areas where there are many modifications and questions. We have also modification other part of the article, please see the text with highlight. I hope my modification has solved your doubts and made you satisfied. Thank you.

Ø 3.2 The third reviewer's comments: Newly modified sections (3.3, discussion conclusion) contain many English mistakes, very long sentences and excessive (sometimes wring) use of punctuation. Please, rather use conjunctions/connectors. All this makes the new added sentences very difficult to read and understand. Please proof-read them.

Response: Thank you for your constructive and helpful suggestion

According to your suggestion, we have proofreading and editing the whole article especially the newly added sentences, and the part of modifications have been highlighted. The content of section3.3, section 4 “Discussion” and section 5 “Conclusion” as below:

  • 3. Effect of coated and uncoated on the proliferation of PSCs

According to the above experimental results, both F10 and DMEM/F12 could significantly promote PSCs proliferation under uncoated conditions. At present, most researchers have coated the culture vessels when conducting cell experiments. Herein, three commonly used coating materials (matrix, rat tail collagen and polylysine) were used to investigate the effects of different basal media on the proliferation of PSCs for 5 days. On the 5th day of culture, the morphology of PSCs cultured in F10 and DMEM/F12 basal medium under both coated and uncoated conditions showed a slender normal morphology (Fig. 3A).With the increase of culture days, the cell viability of DMEM/F12 and F10 all increased significantly no matter coated and uncoated conditions (Fig. 3B). As shown in Fig. 3C,the total number of cells on day 5 was compared with the total number of cells on day 1 to calculate expansion fold. In the F10-based medium, PSCs cultured with the uncoated, rat tail collagen-coated, polylysine-coated and matrix-coated were increased by about 42, 19, 26 and 41 times respectively. The result showed that there was no significant difference in uncoated group and matrix-coated group(P>0.05). In the DMEM/F12-based medium, the expansion fold in uncoated, rat tail collagen-coated, polylysine-coated and matrix-coated group were increased by about 21, 17, 19 and 24 times respectively. The result showed that matrix-coated was the best, uncoated as follow, which have significant difference (P<0.001). To sum up,the effect of F10 was better than DMEM/F12 with or without coating, which was maybe related to the different composition.

In addition, as shown in Fig. 3D, the population doubling time was calculated according to the ratio of cell viability on day 5 to day 1. In the F10-based medium, PSCs cultured with the uncoated, rat tail collagen-coated, polylysine-coated and matrix-coated required about 18, 23, 20 and 18 hours respectively, and this result showed that both uncoated and matrix-coated required the least population doubling time (P>0.05). In the DMEM/F12-based medium, PSCs cultured with the uncoated, rat tail collagen-coated, polylysine-coated and matrix-coated required about 22, 24, 23 and 21 hours respectively. This result showed that uncoated was the best, matrix-coated as follow, which has no significant difference (P>0.05). General speaking, the population doubling time in F10 was shorter than DMEM/F12 no matter coated and uncoated conditions during the same culture period, indicating that cells grew faster in the F10-based medium.

In conclusion, the F10-based medium was more conducive to cell proliferation. Additionally, whether PSCs cultured in F10 or DMEM/F12, the matrix had the best proliferation effect among the three coating materials. Meanwhile, there was no significant difference between matrix-coated and uncoated in F10-based medium, but in the DMEM/F12-based medium, there was a slight difference. This result suggested that differences in medium composition could give rise to differences in cell proliferation on the coated material.”

  • “ Discussion

It is essential to obtain an efficient proliferation medium to shorten cell culture time and produce as many cells as possible within a limited time. Such an approach would be very beneficial to the industrialized production of cultured meat. As some researchers have reported that differentiation potential decrease rapidly, apoptosis and cell senescence increase during long-term in vitro culture. For instance, it has been reported by Wagner et al that mesenchymal stem cell differentiation potential declined and senescence-associated genes were increasingly expressed over time[22]. Qingzi lei et al also have found that the decline in differentiation potential of porcine muscle stem cells was related to culture time rather than cell division number before differentiation[23]. In the previous exploration of proliferation medium for PSCs, it was found that the cell viability and growth rate decreased when PSCs grew to a certain number of generations based on the DMEM-based medium. Interestingly, the cell viability and growth rate increased when the basal medium was replaced by F10 from DMEM. This result demonstrated that different basal media had a great effect on the cells, so two basal media namely F10 and DMEM/F12 were chosen to explore their effects on the growth of muscle satellite cells. Herein, we obtained an efficient proliferation medium based on F10 basal medium that effectively promoted the proliferation of muscle satellite cells from L. crocea in vitro for the first time. By analysis of expansion folds, population doubling time, expression of cell proliferation marker genes Cdk1 and Cdk2 and myogenic differentiation potential, we demonstrated that the effect of F10 was the best. The total cell number of PSCs cultured with DMEM, F10 and DMEM/F12 increased by about 22-fold, 31-fold and 27-fold respectively, and the proliferation of PSCs in the F10-based medium did not affect later differentiation. There were more myotubes that could be formed and the fusion index reached about 20%.

There is no denying that different basal media have diverse functional effects. Related studies have shown that DMEM was used for cell proliferation, whereas, DMEM/F12 was used for cell differentiation and Ham's FI0 which contained nutrients such as vitamins, amino acids, and metabolites was suitable for long-term serum-free cultures [24]. Basal media vary in their composition and content, leading to their utilization in different cultural conditions. Totally speaking, the concentrations of amino acids and vitamins in DMEM were much more abundant than DMEM/F12 and F10. Fayaz and Honaramooz [25] have shown that DMEM sufficiently supported in vitro propagation of gonocytes and somatic cells. Pahlavanneshan, et al. [26] have proved that Ham’s F10 medium provided better conditions for the 2D culture of primary islet cells. Hua, et al. [27] have shown that DMEM/F12 medium was the most suitable media for lower-serum adherent culture among four different media (F12, DMEM/F12, 1640 and DMEM). Wei, et al. [28] have indicated that glutamine was the key molecule to maintain cell growth and survival in culture. In this study, DMEM and DMEM/F12 contained 4 and 2.5mM glutamine respectively, and F10 contained 1.5mM alanyl-glutamine. This composition might be a key factor in promoting PSCs proliferation. Singh, et al. [19] investigated the effects on cell growth by comparing different components of DMEM/F12 medium with DMEM medium, which supporting our findings in PSCs.

Cell adhesion and migration are essential for cell proliferation, some researchers have changed the structure of cell contact media to promote cell adhesion. Du, et al. [29] grafted silk sericin onto the surface of thermoplastic polyurethane (TPU) membrane by -NH2 bridge to build a high-efficiency cell culture membrane, which could promote cell adhesion. Dhania, et al. [30] have shown that different cells exhibited a significant increase in cell viability and attachment when cultured on the porous mats of fabricated polyhydroxyalkanoates blend scaffolds. In this study, we adopted the latter approach by applying a coating material to the cell contact medium to promote cell attachment. Three common coating materials (rat tail collagen, matrix and poly-lysine) were selected. Among them, matrix is widely used to mimic the environment of the extracellular matrix (ECM), which assist in maintaining optimal balance for a series of adjusting cell behaviors, including cell proliferation, cell migration, and cell differentiation [31]. The ECM plays a major role in cell-cell and cell-matrix signaling during normal physiology and disease [32]. The results showed that there was no significant difference in the proliferation of PSCs between coated and uncoated materials, and the effect of matrix was the best among the coated materials. The reason for this phenomenon might be related to species or cell types, and it was indicated that the nutritional components of these three basic media could meet the growth of PSCs. There is no need to provide it with additional external adhesion or migration forces.”

  • “ Conclusions

As we all know, obtaining enough stem cells in vitro is a pivotal step for cultured meat. For this purpose, this study compared with the effects of different basal media on the proliferation and later differentiation of PSCs. Finally, an efficient proliferation medium based on F10 basal medium was obtained. The total cell number of PSCs cultured with F10 increased by about 31-fold, significantly higher than the traditional culture medium DMEM (22-fold). As the same time, in order to promote cell adhesion and migration, three common coating materials (rat tail collagen, matrix and poly-lysine) were selected. Our result showed that there was no significant difference between substrate coated and uncoated on the proliferation of PSCs based in F10 medium. It was also found that the myotubes were washed out and only the ones coated with matrix were intact in the process of differentiation, which might be caused by the strong adhesion ability of matrix. In the matrix-coated condition, the fusion index of F10 reached about 20%. It is indicated that the proliferation of PSCs in the F10-based medium did not affect later differentiation. Taken together, in this study, we obtained an excellent proliferation medium, which could make the rapid proliferation and maintain the stemness of PSCs. Thus, this study provides a basis for large-scale cell expansion and cell culture meat production in the future.”

The above is just to show the areas where there are many modifications and questions. We have also modification other part of the article, please see the text with highlight. I hope my modification has solved your doubts and made you satisfied. Thank you.

It should be noted that one of the graphs in the article is a mistake, we have corrected. The modification as below:

Figure 2. F10 and DMEM/F12 basal medium significantly promoted PSCs proliferation: (E) Quantification of Cdk1, Cdk2 mRNA expression (DMEM high sugar basal medium as a control) The results are presented as mean ± SD. All the experiments were repeated at least three times. *P < 0.05, **P < 0.01, ***P < 0.001.

Round 4

Reviewer 3 Report

The manuscript was revised again and is now ready to be published. Maybe the editorial office can solve minor language issues 

Minor editing (wording especially) required